



**Assimilation of snow water equivalent from AMSR2 and IMS**
**satellite data utilizing the local ensemble transform Kalman filter**
Joonlee lee[1], Myong-In Lee[1]*, Sunlae Tak[1], Eunkyo Seo[2], and Yong-Keun Lee[3]
*[1] Department of Civil, Urban, Earth, and Environmental Engineering, Ulsan National*
*Institute of Science and Technology, Ulsan, Korea*
*[2] Department of Environmental Atmospheric Sciences, Pukyong National University, Busan,*
*South Korea.*
*[3] Earth System Science Interdisciplinary Center, University of Maryland, College Park,*
*U.S.A.*
___________________________________________________________
*Corresponding author: Prof. Myong-In Lee, Department of Urban and Environmental
Engineering, Ulsan National Institute of Science and Technology, 50 UNIST-gil, Ulsan 44919,
Republic of Korea (milee@unist.ac.kr)



## Abstract

The advanced snow data assimilation is developed in this study with satellite remote-sensing retrievals of snow water equivalent(SWE) and snow cover fraction(SCF) utilizing the local ensemble transform Kalman filter based on the Joint U.K. Land Environment Simulator(JULES) land model. The system assimilates SWE from the Advanced Microwave Scanning Radiometer 2(AMSR2) and SCF from the Interactive Multisensor Snow and Ice Mapping System(IMS) during April 2013-2020. The performance is evaluated by the validations with independent data assimilation products derived from in-situ observation.

The baseline model simulation from JULES without satellite data assimilation shows a superior performance in high-latitude regions with heavy snow accumulation, but relatively inferior in the transition regions with less snow and high spatial and temporal variation. Contrastingly, the AMSR2 satellite data exhibit a superior performance in the transition regions, but poor performance in the high latitudes, presumably due to the limitation in the penetrating depth of satellite retrieval. The data assimilation(DA) that combines AMSR2 and IMS satellite data with the JULES model backgrounds demonstrates the positive impacts by reducing uncertainty in both satellite-derived snow data in penetrating deep snow and the model simulations in the transition regions. While DA shows superior performance in most regions, it specifically improves the analysis in the mid-latitude transition regions where the model background errors from the ensemble runs are significantly larger than the observation errors, emphasizing the substantial influence of satellite information. The long-term analysis of snow manifests a pronounced variability in the continental interior at the interannual timescales, which implies large uncertainty in the snow initialization for the sub-seasonal to seasonal predictions of the climate models, potentially degrading prediction skills without satellite snow data assimilation.



## 1. Introduction

Snow plays a crucial role in regulating the water, energy, and carbon exchange between the land surface and atmosphere(e.g., Dutra et al., 2011; Thomas et al., 2016). A snowpack tends to increase surface albedo and soil moisture as the snow melts. It has an impact on the climate system with the water balance by the soil moisture change and the energy balance by albedo variations. In addition to local impacts, the continental snowpack over Eurasia can influence the large scale circulation during winter(e.g., Li and Wang, 2014) or in spring(e.g., Broxton et al., 2017). Especillaly, the Eurasian autumn snow can affect upward-propagating stationary Rossby-wave activity, leading to stratospheric warming and weakening of stratospheric polar vortex and jet stream, which in turn emerges as a negative Arctic oscillation(AO)-like pattern at the surface during winter due to downward propagation through the troposphere. Its impact is shown in both observation and model experiments(e.g., Allen and Zender 2011; Cohen et al. 2007). Therefore, the snow initialization process in climate models is closely related to the improvement of prediction performance.

In the short and medium forecasts, snow is simply prescribed based on climatological values because the forecasts are significantly influenced by the accuracy of the initial atmospheric states in climate models. To extend the accurate prediction to subseasonal to seasonal(S2S) timescales, the atmospheric and the more slowly evolving initial conditions need to be carefully considered. Land initial states such as snow are crucial components in the S2S timescale predictions due to their climatic memory lasting 1-2 months(e.g., Chen et al., 2010). The realistic snow initial states can contribute to improving S2S prediction skills, as proven in several modeling studies(e.g., Orsolini et al., 2013; Li et al., 2019).

Snow states are generally provided from in-situ observations data, remote-sensing retrievals from satellites, or numerical models such as the land surface model(LSM) operated based on the observed atmospheric variables. In the case of the in-situ data, the primary source



of snow depth(SD) is obtained from surface synoptic observations(SYNOP). These
observations are provided almost in real-time through the global telecommunication
system(GTS). In addition to SYNOP, there are some regional snow measurement networks.
For instance, the snowpack telemetry(SNOTEL) network collects data on SD over 900
automated observation points located in the western United States, and the National Oceanic
and Atmospheric Administration (NOAA) Cooperative Observer Program gathers SD data in
North American region. Nevertheless, data collected from these national networks cannot be
utilized in the almost real-time GTS. Directly measured in-situ data provide the most reliable
snow information but have relatively coarse temporal and spatial resolutions over the limited
area because of spatial heterogeneities of snow(Helmert et al., 2018). Recently, in order to
obtain high-resolution and high-quality snow water equivalent(SWE) analysis, artificial
intelligence(AI) such as long short-term memory(LSTM) has been utilized with the given
meteorological conditions from SNOTEL observations as input data, but it is still insufficient
to cover the entire globe(Meyal et al., 2020).
Satellite-derived observations using conical scanning microwave instruments may provide
spatially consistent data coverage across the globe. Cho et al.(2017) showed the SWE retrieval
results from two passive microwave sensors, the advanced microwave scanning radiometer
2(AMSR2) and the special sensor microwave imager sounder(SSMIS). However, the
algorithms for SWE retrieval exhibit a degree of sensitivity to a variety of parameters such as
snow liquid water content and snow grain size distribution(De Rosnay et al., 2014). Hence,
satellite-based SWE data still have limitations in accuracy, especially under deep snow
conditions due to the restrictions in penetration depth(Gan et al., 2021). On the other hand,
satellite retrieval can estimate snow cover accurately under clear sky conditions (Brubaker et
al., 2009). The moderate resolution imaging spectroradiometer(MODIS) instrument observes
daily snow cover, while a multi-satellite-based interactive multi-sensor snow and ice mapping





system(IMS) provided by the United States National Snow and Ice Data Center produces the
snow cover by combining in-situ observations and satellite data from microwave, infrared, and
visible sensors.

Model simulations can cover complete spatiotemporal resolution but involve potentially

large uncertainties due to the deficiencies in the physical parameterizations and meteorological
forcing data(Dirmeyer et al., 2006; Seo et al., 2020). To reduce the uncertainties from model
simulations, previous studies have used satellite-based snow cover and in-situ observation such
as SYNOP SD available on the GTS, in conjunction with the model simulation(e.g., Brasnett,
1999; Dee et al., 2011; Meng et al., 2012; Pullen et al., 2011; De Rosnay et al., 2014). For
example, the snow analysis for the Canadian Meteorological Center(CMC) utilizes a 2-
dimensional optimal interpolation(2D-OI) scheme with in-situ observations and the outputs
from a simple snow model(Brown et al., 2003). The National Centers for Environmental
Prediction (NCEP) climate forecast system reanalysis(CFSR) combines the IMS as satellite-
based snow cover retrieval and the outputs from the global SD model of the Air Force Weather
Agency(Meng et al., 2012). At the European Center for Medium Weather Forecast (ECMWF),
the ECMWF reanalysis (ERA)-Interim and ERA5 for the snow analysis employ a Cressman
interpolation and 2D-OI, respectively, with the IMS, in-situ observation, and the results from
a land surface model(Dee et al. 2011; De Rosnay et al., 2014). The Japanese 55-year
Reanalysis(JRA55) also utilizes the 2D-OI with in-situ observation, satellite-based snow cover
from SSMIS, and the results from an LSM(Kobayashi et al., 2015).

The most commonly employed approach to obtain reasonable estimates of land initial states

for predictions is running atmospheric general circulation models(AGCMs; Pullen et al., 2011)
or offline mode of LSMs with observed atmospheric conditions(Dirmeyer et al., 2006). Climate
prediction systems in operational centers such as the Meteorological Office(Met Office) (Met
Office) in the United Kingdom and the Korean Meteorological Administration(KMA) conduct



123 the snow initialization by utilizing the results of the operational global unified model(UM) and

124 the IMS snow cover(Pullen et al., 2011). The initialization at NCEP also performs a similar

125 approach using input data combined from IMS snow cover and results from the global SD

126 model(SNODEP; Meng et al., 2012). Furthermore, the snow initialization of ECMWF employs

127 optimal interpolation with a combination of results from the LSM, IMS snow cover, and in-

128 situ observation from SYNOP and national networks available on the GTS. However, in areas

129 where ground observations are not available, the results of the snow model are relied upon,

130 which still exists significant uncertainty in snow accumulation because of uncertainties in the

131 atmospheric forcing and imperfect model parameterizations(Boone et al., 2004; Essery et al.,

132 2009). It would be useful to to accurately initialize the snow amount including vertical depth,

133 which is more important in estimating energy and water budgets, by using the satellite-derived

134 snow amounts with comparatively uniform spatial and temporal resolution.

135  However, the SWE retrievals from satellites still have considerable uncertainties(De

136 Lannoy et al., 2010; Dawson et al., 2018), which can arise from vegetation and terrain

137 interference, sensor signal saturation, snowfall amount, and simplifications in the underlying

138 assumptions of the retrieval algorithms(Liu et al., 2015). In particular, a region with heavy

139 snow accumulation leads to a significant underestimation of SWE due to the limitations in

140 penetration depth from satellites(Gan et al., 2021). For this reason, satellite-derived SWE is not

141 employed in the land initialization process. Nevertheless, the SWE retrieval shows important

142 advantages such as high performance in shallow snow areas with temporal and spatial

143 homogeneity(Gan et al., 2021). In previous studies, various approaches have been attempted

144 to improve SWE product performance, such as combining satellite-derived SWE with ground

145 observations(Pulliainen et al., 2020), different satellite data sets(Gan et al., 2021), simple snow

146 models(Dziubanski and Franz, 2016), or LSMs(Kwon et al., 2017). For instance, Kumar et

147 al.(2019) show the improvement of the SD estimation over the contiguous United States by





assimilating satellite snow SD into the Noah LSM, indicating that these model-based products
are generally superior to stand-alone satellite-based SWE retrievals. Thus, a globally advanced
snow initialization such as data assimilation using satellite snow amount is ideal for providing
realistic snow initial states related to S2S prediction skills.
Therefore, the purpose of this study is to develop an advanced snow assimilation system
utilizing the Local Ensemble Transform Kalman Filter(LETKF) with satellite-derived
observations of SWE, IMS snow cover, as well as the Joint U.K. Land Environment
Simulator(JULES). In this context, our focus is on SWE rather than SD, because the former
can be used directly for hydrological analysis and initial states of the model(Gan et al., 2021).
From this novel assimilation system, we endeavor to achieve the following objectives. The
primary aim is to assess the enhancement in SWE performance through the assimilation with
satellite remote sensing data. The satellite data show high performance in the transition regions
with climatologically shallow conditions, termed by Koster et al.(2004) as "hot spots" of
atmosphere-land coupling. The second goal is to reveal the reason for skill improvement with
the snow data assimilation, based on the Kalman gain analysis that measures the ratio of the
model errors with respect to the observation errors. From these perspectives, it would be
possible to know how much the satellite has affected the transition regions, and how the
assimilation system deals with the regions of deep snow accumulation where the satellite has
difficulty in accurate retrieval. The final goal is to evaluate the advantages of assimilating
satellite retrievals in extreme high-temperature events, specifically over Eurasia in April 2020.
In this regard, we expect that the data assimilation of satellite-derived snow information can be
an alternative to produce optimal snow initial states for improving the S2S prediction skill in
the climate models.

**2. Data and model**



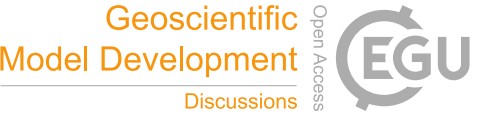

### 2.1. Satellite data


The snow information including snow cover and SWE can be derived from satellite
measurements offering global coverage and high temporal as well as spatial resolution. For
data assimilation, this study uses SWE calculated from brightness temperature measurements
obtained by the AMSR2 on board the Japanese Aerospace Exploration Agency (JAXA) global
change observation mission-water(GCOM-W) satellite. This AMSR2 Unified Level-3(L3)
dataset offers daily estimation of SWE at 25 km resolutions from July 2012 to the present.
AMSR2 has a sensor designed to detect microwave radiation naturally emitted from the surface
and atmosphere, employing six frequency bands ranging from 6.9 to 89 GHz. Through this
conical scanning mechanism, AMSR2 can acquire day and night datasets with nearly constant
spatial resolution over more than 99% of the global coverage every two days. Comprehensive
explanations of AMSR2 characteristics are available in Imaoka et al.(2010). AMSR2 is
selected for the assimilation because it produces more skilled results by assimilating data from
modern sensors(e.g., AMSR2) compared to data from conventional sensors(e.g., AMSR-
E)(Cho et al., 2017).

### 2.2. Reference data for SWE and SCF


The CMC daily estimated SWE is used for verification. The SWE data is processed using
statistical interpolation between a background field derived from a simple snow model and in-
situ daily SD(Brown and Brasnett, 2010). In detail, this dataset utilizes optimal interpolation
methods to acquire spatial SD from the in-situ data, involving SYNOP, special aviation reports
from the World Meteorological Organization(WMO), and meteorological aviation
reports(METAR). In areas with scant in-situ data, a simple snow accumulation and melt model
is employed to create an optimal interpolation that estimates snowmelt and snowfall worldwide,
assuming the persistence of the snowpack mass between snowfall and melting



events(Brasnett, 1999). Although the average elevation of snow measurement stations used in
CMC is biased toward low elevations(< 400m), leading to a potential negative bias at high
elevations, the CMC dataset is often considered the premier snow analysis accessible in the
Northern Hemisphere(Su et al. 2010) and has still been widely used to evaluate model
outputs(e.g., Reichle et al., 2011; Reichle et al., 2017; Toure et al, 2018). Therefore, the SWE
of CMC produced without the satellite-derived data is selected for verification as an
independent dataset for evaluating the assimilated analysis with remote sensing snow retrievals.
Since only daily SD analysis is provided in CMC, it is converted to daily SWE based on the
snow bulk density methods(e.g., Sturm et al., 2010). It is available from 12 March 1998 to the
present and offers comprehensive coverage of the entire Northern Hemisphere with a
horizontal resolution of 24 km. The SWE of CMC at its native horizontal resolution is
interpolated onto the LSM grid through local area averaging.

The widely used multisensor–derived snow cover is IMS(e.g., Ramsay 1998; Helfrich et

al., 2007) produced by NOAA the National Environmental Satellite Data and Information
Service(NESDIS) for the Northern Hemisphere from February 2004 to the present at 4 km
resolutions. This dataset is generated using various data products, including multi-satellite
images and in-situ observations(U.S. National Ice Center, 2008). Since IMS provides binary(0:
no snow or 1: snow covered) snow cover information, we transform the IMS snow cover at 4
km grids to the snow cover fraction(SCF) within a 50-km LSM grid by counting the snow pixel
number with a value of 1. A 50-km LSM grid is declared as snow-covered when more than 50%
of the 4km pixels within the grid are covered with snow. In this study, the IMS-based SCF is
employed to mask the SWE, considering the higher reliability of IMS data (e.g., Brown et al.,

2014).


**2.3. JULES LSM**





This study utilizes the JULES LSM from the Met Office(Best et al., 2011), a component
land model of the global seasonal forecasting system version 6(GloSea6) global, fully-coupled
atmosphere, ocean, land, and sea-ice model. The surface types(or snow tiles) in the JULES
LSM consist of four non-vegetated types: urban, land-ice, inland water, and bare soil, as well
as five vegetation functional types: C3 temperate grass, needleleaf trees, shrubs, C4 tropical
grass, and broadleaf trees. For each surface tile, a separate energy balance is computed, and the
average energy balance in the grid cells is determined by applying weights to the values of each
surface tile. Two schemes are used within JULES to represent surface snow. The simple
method involves a zero-layer approach, which modifies the top soil level without using explicit
model layers to represent snow processes. The other is the multi-layer approach which is more
comprehensive. In the case of vegetated surfaces, snow can be separated into ground snow and
canopy snow or stored in a single effective reservoir. As both the zero-layer and multi-layer
snow models provide similar results under various conditions(e.g., Best et al., 2011), this study
used the zero-layer snow model with constant thermal conductivity and density for snow.
Although the heat capacity of snow is ignored, the bulk thermal conductivity in the surface
layer is reduced as the thermal conductivity of snow differs from that of the soil and the layer
thickness increases. As long as snow persists on the ground, the skin temperature cannot exceed
0°C, yet the heat flux utilized for melting the snow is diagnosed through the residual surface
energy balance. The melted water is immediately drained from the snow, divided into runoff
and soil infiltration, and liquid water is not stored or frozen in the snow. A detailed description
of the energy and water cycling in the JULES LSM can be referenced in Best et al.(2011).
The prognostic variables(e.g., SWE) in the LSM are determined by meteorological forcing
variables such as 2-m air temperature, humidity, 10-m wind speed, precipitation, surface
pressure, and radiative fluxes. The 3-hourly, JRA55 reanalysis at 0.56° spatial resolution is
employed for the meteorological forcing variables, which is linearly interpolated to a 50 km





resolution of the LSM. The model background error needed for data assimilation is estimated
by JULES ensemble runs with perturbed initial and boundary conditions. Following the
previous studies(Reichle et al., 2008; Seo et al., 2021), meteorological forcing variables are
perturbed due to randomness, especially precipitation, downward shortwave, and downward
longwave. Perturbations are applied using additive adjustments assuming a normal distribution
for longwave radiation and multiplicative adjustments following a log-normal distribution for
shortwave radiation and precipitation. Here, the ensemble means of additional and
multiplicative perturbations are zero and one, respectively. The relationship between disturbed
precipitation and radiative flux ensures the physical consistency among atmospheric forcing
variables(Reichle et al., 2008). For instance, a negative anomaly in precipitation and downward
longwave-radiation is statistically linked to a positive anomaly of downward shortwave-
radiation. Detailed explanations regarding the perturbation of atmospheric forcings can be
found in Reichle et al.(2008).



## 3. Methodology

### 3.1. Bias correction

The discrepancies in SWE between remote sensing and LSMs are often caused by uncertainties in the model physics and meteorological forcing data. These differences can lead to significant biases in the variance and mean of SWE between model simulations and satellite remote-sensing retrievals, and such biases can result in poor performance. In previous studies(e.g., Reichle and Koster, 2004; Seo et al., 2021), a scaling method of the nonlinear cumulative distribution function(CDF) matching is used to account for the systematic bias of soil moisture in the model backgrounds. However, in this study, it is difficult to apply it as the CDF distribution of SWE could not be clearly simulated due to the insufficient sample size. To address this issue, we attempted to apply a simple and effective standard normal deviation scaling to satellite-derived SWE. Based on the climatology and standard deviation for the model and remote sensing retrievals, the scaled SWE($O_{new}$) from the satellite can be derived from the following relation:

$$O_{new} = \left(\frac{O - \bar{O}}{\sigma_o} \times \sigma_m\right) + \bar{M} \tag{1}$$

, where $\bar{O}(\sigma_o)$ and $\bar{M}(\sigma_m)$ indicate climatology(standard deviation) of remote sensing retrievals and the model, respectively. This approach has been widely utilized in observation-based land initialization and has proven to be effective(e.g., Koster et al., 2011; Jeong et al., 2013).

### 3.2. Snow assimilation method

The snow assimilation is conducted based on the LETKF(e.g., Hunt et al., 2007), which is



utilized to combine satellite remote-sensing retrievals with the LSM model outputs(a.k.a.
backgrounds) to produce a snow analysis. LETKF is a powerful data assimilation method and
has several advantages over other methods. First, LETKF can efficiently handle large datasets
and high-dimensional state variables by localizing the covariance matrix. This offers efficiency
in parallel computing, making it suitable for real-time forecasting and high-resolution data
assimilation. Secondly, the method utilizes model simulation ensembles to capture the
uncertainty in the initial states and model errors, which allows for a better representation of the
true probability distribution of the state variables that vary in time and space. Third, LETKF
applies an adaptive inflation scheme, which adjusts the ensemble spread to account for the
observational and model errors, ensuring that the uncertainty estimates are realistic and not
underestimated nor overestimated. In LETKF, the analysis states($X^a$) are obtained by
$$X^a = \bar{X}^a + \delta X^a \tag{2}$$

, where $\bar{X}^a$ and $\delta X^a$ are the matrices of analysis ensemble means and perturbations,
respectively. They are defined by
$$\bar{X}^a = \bar{X}^f + \delta\tilde{x}^a \tag{3}$$

$$\delta X^a = \delta X^f [(K-1)\tilde{P}^a]^{1/2}. \tag{4}$$

Here, the analysis ensemble means($\bar{X}^a$) is determined by gathering the analysis increment($\delta\tilde{x}^a$)
to the model ensemble mean($\bar{X}^f$) produced by the JULES land surface model. The analysis
ensemble perturbation($\delta X^a$) is computed considering the model perturbation($\delta X^f$), the
number of model ensembles($K$), and the analysis error covariance($\tilde{P}^a$) in the ensemble space.
The analysis increment($\delta\tilde{x}^a$) is acquired by considering the difference between the SWE of
AMSR2 used as observation and the model ensembles produced by the JULES LSM and
determined by
$$\delta\tilde{x}^a = \delta X^f \tilde{P}^a \delta Y^T R^{-1}(y^o - \overline{H(X^f)}), \text{ and} \tag{5}$$

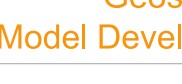
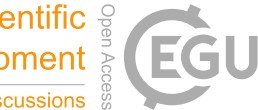

$$\tilde{P}^a = [\frac{(K-1)I}{\rho} + \delta Y^T R^{-1} \delta Y]^{-1}. \tag{6}$$


It consists of the model ensemble perturbation($\delta X^f$), the analysis error covariance($\tilde{P}^a$),
observation error covariance($R$), model ensemble perturbation in the observation grid($\delta Y$), and
observation innovation $\left(y^o - \overline{H(X^f)}\right)$ derived from the difference between the model
ensemble in the observation grid($\overline{H(X^f)}$) and the observation($y^o$). Here, $H$ represents the
observation operator, projecting the modeled snow background onto the satellite observation
locations using bilinear interpolation. $\rho$ denotes the covariance inflation factor for the $\tilde{P}^a$,
aiding in preventing underestimation of the covariance. This study applies multiplication-based
20% inflation($\rho$) for the ensemble spread derived from 24 member ensembles. Therefore, the
final analysis state($X^a$) is written as

$$X^a = \bar{X}^f + \delta X^f \left[\tilde{P}^a \delta Y^T R^{-1}\left(y^o - \overline{H(X^f)}\right) + [(K-1)\tilde{P}^a]^{\frac{1}{2}}\right]. \tag{7}$$


This approach involves the weight function($w(d_j)$) for the covariance localization within
the local patch centered at the analysis grid(e.g., Houtekamer and Mitchell, 2001; Hamill et al.,
2001). This function assigns larger errors to observations located farther away from the center
of the local patch, as proposed by Miyoshi and Yamane(2007), depending on the Gaussian
function as

$$w(d_j) = e^{\frac{-d_j^2}{2\sigma^2}} \tag{8}$$


where σ denotes a parameter of the localization length scale and $d_j$ indicates the distance of
j-th observed value from the center of each local patch. In this study, the horizontal local patch
size and the localization length scale parameters are defined as 150 km and 30 km(Table 1),
respectively. Detailed information about the LETKF algorithm and its implementation can be
referenced in Hunt et al.(2007) and Shlyaeva et al.(2013).



### 3.3. Snow data assimilation design

This study conducts the advanced snow data assimilation experiment at a daily cycle based on LETKF with the satellite data and the JULES LSM model outputs driven by 3-hourly JRA55 reanalysis atmospheric forcing. The snow assimilation processes are illustrated in Fig. 1, with a more detailed description in Table 1. Since data assimilation is conducted by considering the error of SWE in both the model and the observation, it is important to accurately understand the observation and model errors to improve the performance of data assimilation. The experiment calculates the model error from the 24 ensemble member spreads generated by perturbing atmospheric forcings such as longwave radiation, shortwave radiation, and precipitation in JULES LSM, as provided in section 2.3. The observation error is conservatively prescribed as 10% of AMSR2 SWE for each grid compared to the previous study(Lee et al., 2015), because it usually increases during the snow accumulation period with the developing deep snowpack(Foster et al., 2005; Cho et al., 2017). Here, the bias-corrected AMSR2 satellite data as described in section 3.1 is used as the observation data, and the updated analysis state($X^a$) through data assimilation becomes a new initial state for the next integration in JULES LSM(Fig. 1). In addition, the analysis state of this method is calculated based on the IMS snow cover fraction as a reference in the following way(Fig. 1); where the SCF of IMS is zero, the snow amount analysis is set to zero, and in other cases, it is derived from data assimilation. The reason for this is due to the importance of the presence or absence of snow in the climate system, as well as the high reliability of the IMS data. A background experiment of JULES LSM without satellite data assimilation as a baseline(referred to hereafter as "Openloop") is also achieved by employing the same ensemble perturbations, thereby measuring the skill improvement from the snow analysis state through the assimilation of satellite-derived SWE and IMS SCF from satellite and surface observations(referred to hereafter as "DA"). All experiments are conducted in April from 2013 to 2020, which is one





of the months with low snow performance in the LSM when the snow begins to melt in the

Northern Hemisphere(e.g., Toure et al., 2018; You et al., 2020).



## 4. Results

### 4.1. Skill Verification

Figure 2 displays the climatological-mean SCF from the IMS multi-satellite data(Brown et al., 2014) and the differences from AMSR2, Openloop, JRA55, and DA for April 2013-2020. April is a season when the accumulated snow during the cold season begins to melt. This study defines the transitional region with a climatological-mean SWE of less than 16 mm as in previous studies(e.g., Gan et al., 2021), the boundary of these transition regions is represented by the black lines in Fig. 2. The transitional regions exhibit large variability in space and time, and they are mainly located at mid-latitudes. The SCF climatology patterns show negligible differences in high latitudes of heavy snow accumulation but noticeable differences in the transitional mid-latitude regions of less snow. SCF from JRA55 tends to be underestimated compared to IMS, whereas AMSR2 and Openloop tend to overestimate. There is a clear difference in SCF between AMSR2 and IMS satellite data. This study gives more credibility to IMS than AMSR2, as the former is based on multiple satellite data sources. As we used the IMS SCF to define the snow region to be assimilated by AMSR2 SWE, it is natural that DA shows better consistency with IMS and reduces overestimation biases in Openloop. Quantitatively, the root mean square differences(accuracy) for AMSR2, Openloop, JRA55, and DA with(from) IMS are 0.23(0.91), 0.18(0.91), 0.13(0.93), and 0.13(0.97), respectively, showing the best consistency in DA.

The SWE climatology from AMSR2, Openloop, JRA55, and DA is also compared with CMC as a reference in Fig. 3. The SWE derived from AMSR2 shows a significant underestimation compared to CMC, particularly in the regions with heavy snow accumulation at high latitudes. This is presumed to be due to limitations in satellite sensors detecting the depth of snow(Gan et al., 2021). On the other hand, the climatological SWEs from Openloop and DA exhibit higher correspondence to CMC, even higher than JRA55. Specifically, DA



demonstrates a higher agreement with CMC. Quantitatively, the pattern correlation
coefficients(root mean square differences) for AMSR2, Openloop, JRA55, and DA with(from)
CMC are 0.63(80.7 kg/m$^2$), 0.80(50.1 kg/m$^2$), 0.60(100.8 kg/m$^2$), and 0.80(49.9 kg/m$^2$),
respectively. DA with snow data assimilation displays the highest correlation and the smallest
root mean square difference among the datasets, indicating the benefit of assimilating the
AMSR2 SWE despite the relatively lower performance of the satellite data itself.
Next, we compare the temporal variation of SWE as measured by the Spearman rank
correlation coefficient with CMC, which is regarded as more appropriate than the Pearson
correlation coefficient for describing nonlinear variables such as snow in both time and space.
Figure 4 compares the distribution of correlation skills from AMSR2, Openloop, JRA55, and
DA. Openloop has a high performance in regions with heavy snow accumulation but relatively
low performance in transition regions with significant snow changes. In contrast, the results
from the AMSR2 satellite data represent poor performance in high-latitude areas with heavy
snow accumulation but high performance in transitional regions, consistent with the previous
studies(Gan et al., 2021). DA shows high performance not only in high-latitude areas with
heavy snow accumulation but also in transition regions. Even compared to JRA55 used as the
atmospheric forcing, DA performs better in temporal variation. The quantitative results in the
correlation in the Northern Hemisphere over 40ºN(the transition region) are 0.41(0.54) for
AMSR2, 0.61(0.48) for Openloop, 0.58(0.58) for JRA55, and 0.67(0.61) for DA, respectively.
The findings indicate that satellite retrievals offer additional value in capturing temporal
variations through data assimilation.
The performance improvement by DA is also evident in the zonally-averaged correlation
coefficient shown in Fig. 5. The AMSR2 satellite data shows higher performance than
Openloop in the transition region around latitude 45 ºN-55 ºN, although performance sharply
decreases with increasing snow accumulation. Openloop indicates gradually increasing





performance as the latitude increases, with the highest performance at around 60ºN. DA

denotes superior performance across the Northern Hemisphere, especially in the mid-latitude

transition region than AMSR2 or JRA55. An exception is for 35-40ºN in the Tibetan Plateau,

where JRA55 used in-situ observations. The results suggest that the developed snow data

assimilation system represents well not only the transitional regions but also the satellite-

limited regions with heavy snow.

Figure 6 presents the Spearman rank correlation depending on the SWE amount in the

Northern Hemisphere. AMSR2 exhibits higher performance than Openloop for SWE up to 16

mm. However, the performance of AMSR2 sharply declines beyond that threshold, and

Openloop shows a better performance. Consistent with the results illustrated in Figs. 4 and 5,

DA demonstrates superior performance compared to others. Note that DA performs

significantly better in the transition region of less than 16 mm of SWE. Considering that the

area below 16 mm of SWE accounts for approximately 53% of the entire area of the Northern

Hemisphere(as shown in the pie chart in Fig. 6), the data assimilation impact is identifiable,

and it can contribute substantially to the increase in the prediction skill through improving the

simulation of the albedo changes and surface energy balance.

This study conducted a further sensitivity test to investigate the influence of incorporating

IMS snow cover in snow assimilation. Figure 7 compares the correlation differences between

Openloop and the data assimilation result employing both AMSR2 and IMS(DA), as well as

the data assimilation result utilizing solely AMSR2 and excluding IMS(hereafter referred to as

DA_AMSR2). The results obtained from the snow assimilation show the improvements in the

transitional regions where AMSR2 denotes a better agreement with the observations compared

to Openloop. Notably, the skill is enhanced significantly in DA by incorporating the IMS SCF.

There are exceptional areas where DA performs inferior to Openloop, which are associated

with the differences in SCF between IMS and CMC. Moreover, the performance of SWE





improves even when only AMSR2 is used, but incorporating IMS leads to a substantial
improvement in the transitional regions. This implies that IMS has a positive influence on the
snow data assimilation.


**4.2 Kalman gain analysis**
In order to better understand the skill enhancement through snow assimilation of satellite
data, this section examines the Kalman gain, which represents the weights of the assimilated
observations in the analysis update of LETKF. Figure 8 illustrates the spatial distribution of
observation error, model background error, and the Kalman gain. A high value of the Kalman
gain denotes that the assimilated result is closer to the AMSR2 observation than the model
background. The Kalman gain is large when the model error becomes large, or the observation
error is small. As this study specifies the observation error as a conservative 10% of SWE
compared to the previous study(Lee et al., 2015), the observation error basically follows the
distribution similar to the climatological-mean values. The background errors, originating from
the 24 ensemble members, have higher values in high-latitude regions and mid-latitude regions.
Data assimilation methods such as LETKF used in this study often face challenges in accurately
representing background errors when the ensemble spread is insufficient. Generally, the
magnitude of ensemble spread is frequently compared to the root mean square error(RMSE).
The background error in this study demonstrates a sufficiently valid magnitude in comparison
with the RMSE, as illustrated in SFig. 1, indicating that it is well estimated. In the spatial
distribution of Kalman gain in Fig. 8c, significant performance improvement is observed in
transition regions, where Kalman gains exhibit larger values. However, in high-latitude areas
with substantial snow accumulation, there is a tendency for Kalman gain to have lower values.
These findings agree well with the bar graph in Fig. 9, which illustrates the Kalman gain as a





function of SWE amount. In the region encompassing the transition region with SWE amounts
below 20 mm, the Kalman gain displays the highest values, particularly exceeding 0.8. As the
SWE amount increases, the Kalman gain decreases, with a significant decline observed when
the SWE amount reaches 80-100 mm or higher. Furthermore, in the areas where DA denotes
improved skill compared to Openloop, the Kalman gain shows values generally above 0.7. In
contrast, relatively lower values below 0.5 are observed in the areas with decreased skill. This
indicates that in the dominant areas of performance improvement, including the transition
region, the background error is significantly larger than the observation error, emphasizing the
substantial influence of observations in data assimilation. It is found that accurate remote
sensing retrievals are well reflected in regions with high uncertainty in the LSM through the
snow data assimilation system, leading to performance improvement.


**4.3 Validation of the SWE for the extreme event**

In April 2020, Siberia experienced a record-breaking heatwave with the highest observed

average temperature. This section investigates the potential benefits of snow assimilation using
satellite data for the case of the 2020 Siberian heatwave. Previous studies have identified the
strong polar vortex accompanied by the AO amplification during winter as a major cause of
the cold Eurasian(Overland and Wang, 2021). Additionally, it has been revealed that the
occurrence of high temperatures in the Siberian region is found to be closely related to the
development of large-scale atmospheric waves in the upper atmosphere of the Eurasian region,
indicating a significant influence on the strengthened land-atmosphere interaction in recent
years. As a result, remarkable snow melting occurred due to the high surface temperature over
the Siberian region in April 2020, leading to extremely low values of SWE and SCF as depicted
in SFig. 2. This is consistent with previous studies reporting a significant snow depletion in





2020 in the region(Gloege et al., 2022). Especially, as shown in Fig. 10, significant negative
anomalies in SWE and SCF are predominant over the transition region. With a substantial snow
melting, it increases the sensible heat flux to the atmosphere, thereby strengthening the upper-
level waves by enhanced atmosphere-land interaction, leading to further intensification of
heatwaves. This implies the importance of realistic snow initial states in the global coupled
model forecasts. For the Siberian region with extreme high-temperature events marked by the
red box in Fig. 10, DA shows a better agreement with the extremely dry snow conditions,
especially in the transitional region, compared to the Openloop. These results are evident when
considering the observation-to-model ratio in that region. The percentage of CMC(IMS) is
83%(78%) for Openloop and 93%(89%) for DA, indicating that DA with snow data
assimilation based on satellite data produces more significant changes in snow in comparison
with Openloop. Similarly to the 2020 case, we obtained another significant case in 2014
compared to Openloop, as shown in SFig. 3. Such extremely dry snow conditions can provide
significant heatwave events in the following months.










## 5. Conclusions

The advanced snow data assimilation is developed in this study with the LETKF data assimilation method based on the JULES LSM. The system assimilates snow retrievals from AMSR2 and IMS remote sensing observations. This study showed that the satellite-derived snow data has limitations in penetrating deep snow, and exhibited much discrepancy from the snow obtained from the Openloop LSM simulations. The snow assimilation framework developed in this study proves the beneficial impacts of using satellite snow data, maintaining better analysis quality both in the regions with low satellite data quality and the high satellite data quality by dynamically balancing the errors from the satellite observations and the model background forecasts. It is found that the simulation from Openloop as a baseline shows superior performance in high-latitude regions with heavy snow accumulation but relatively inferior performance in transition regions with significant snow changes. Contrastingly, the results of the AMSR2 satellite data represent poor performance in high-latitude regions but exhibit good performance in transition regions. AMSR2 demonstrates higher performance than Openloop up to 16 mm of SWE, but beyond that threshold, the skill of AMSR2 sharply declines while Openloop shows better performance. DA with snow data assimilation consistently performs better in the climatological-mean pattern and temporal variation compared to other results. Notably, the snow assimilation system in this study reflects well the errors and advantages of land surface models and satellite-derived data, controlling not only the transition regions but also the satellite-limited regions with heavy snow.

The significant improvement of SWE data assimilation is primarily observed in the transition regions of less than 16 mm, which accounts for approximately 53% of the entire areas of the Northern Hemisphere. A sensitivity test also revealed that the use of IMS SCF led to a substantial improvement in the transitional regions, in addition to the use of AMSR2 SWE. The sources contributing to the skill improvement of SWE in the snow assimilation system can



be explained through Kalman gain analysis, measuring the relative importance of observations
given the model background errors. Higher Kalman gains values above 0.7 were observed in
the transition regions, whereas they decreased below 0.5 in the high latitudes with heavy snow
accumulation. It found that in the dominant areas of performance improvement, including the
transition region, the background error is significantly larger than the observation error,
emphasizing the substantial influence of observations in the snow assimilation process.

In the case of the Siberian heatwave, remarkable snow melting occurred due to high surface

temperature over the Siberian region in April 2020. It resulted in extremely low values of SWE
and SCF, leading to a further intensification of the heatwave. The SWE anomalies from the
snow data assimilation with the AMSR2 satellite showed significant changes in snow that
seemed to better explain the heatwave episode than the Openloop.

The quality of the observation is crucial in the data assimilation system. Satellite-derived

snow cover exhibits a significantly higher accuracy compared to other data sources, while SWE
has restricted performance due to the limitations of penetration depth by satellite sensors and
relies heavily on estimation algorithms. Due to these problems, most previous studies and
operational centers primarily depend on satellite-derived snow cover for snow initialization.
However, the findings from this study highlighted the beneficial impacts of using satellite-
derived SWE, particularly in the rapidly changing transition areas, to find out which variable
is more important in closing surface energy and water balance changed by snow. Nevertheless,
areas of significance in large-scale circulation, such as the Tibetan region, which experiences
significant uncertainty and degraded performance in satellite data, do not exhibit substantial
data assimilation effects. As the performance of SWE derived from various satellites continues
to advance, these issues will be discussed more.

Improved snow estimates from the snow assimilation system can enhance the initialization

of climate models used in most of the seasonal forecast operation centers. As snow significantly



influences energy and water balance at the atmosphere-land boundary, this approach allows for
a more accurate prediction of atmospheric conditions by realistically representing atmosphere-
land interactions. Specifically, this applies to transitional regions where the reliability of snow
estimation performance through model simulations is compromised. The long-term analysis of
snow manifests a pronounced variability in the continental interior at the interannual timescales,
potentially improving the prediction of extreme heatwave events by global couple models. This
study used the gridded CMC data as a validation reference, which is based on in-situ
observations. Despite much uncertainty and limitations of this dataset, we expect to obtain
comparable conclusions to this study through comparisons with other independent,
observation-based datasets.






**Key words**

Snow data assimilation, AMSR2, LETKF, snow water equivalent, JULES LSM

*Data availability.*

The AMSR2 SWE and IMS SC were obtained from https://n5eil01u.ecs.nsidc.org/AMSA/AU_DySno.001/ and https://noaadata.apps.nsidc.org/NOAA/G02156/, respectively. The CMC SWE was collected from https://daacdata.apps.nsidc.org/pub/DATASETS/nsidc0447_CMC_snow_depth_v01/. The snow-assimilated results and land surface variables from the LSM offline simulation may be requested from the authors.

*Author contributions.*

LJL conceived the project, designed the study, developed the snow assimilation system, wrote the paper, and made the figures. LMI provided advice on the methods, project design, and review and editing of the manuscript. TSL helped with the experiment with the land surface model. SEK helped with the data assimilation method based on LETKF. LYK provided advice on snow satellite data and the sensitivity methods. All authors contributed to the writing of the paper by providing comments and feedback.

*Competing interests*.

The contact author has declared that none of the authors has any competing interests.



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





**Table 1**. Description of the land surface model, the data used, and assimilation experiment
designs.

| | INFORMATION | REFERENCES |
|---|---|---|
| Land Surface Model | JULES | Best et al., (2011) |
| Atmospheric Forcing | 3-hourly JRA-55 reanalysis | Kobayashi et al., (2015) |
| Snow Observation | AMSR2 & IMS | Imaoka et al., (2010) |
| | | Ramsay (1998) |
| | | Helfrich et al., (2007) |
| Data Assimilation scheme | Local Ensemble Transform Kalman Filter (LETKF) | Hunt et al., (2007) |
| | | Miyoshi and Yamane, (2007) |
| Resolution (km) | 0.5° ×0.5° (~ 50) | |
| | 1-day DA cycle | |
| Localization patch size (km) | 3×3 (150), σ =30 | |
| Ensemble sizes | 24 | |
| Experiment period | 2013-2020, APR | |








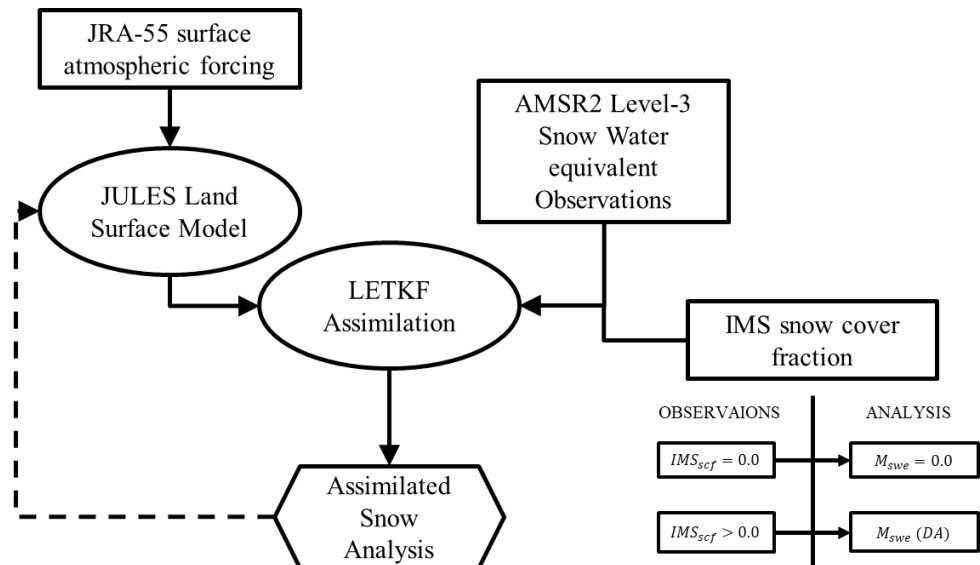


**Figure 1**. Schematic diagram of the snow assimilation system with satellite-derived

observations and the land surface model outputs.





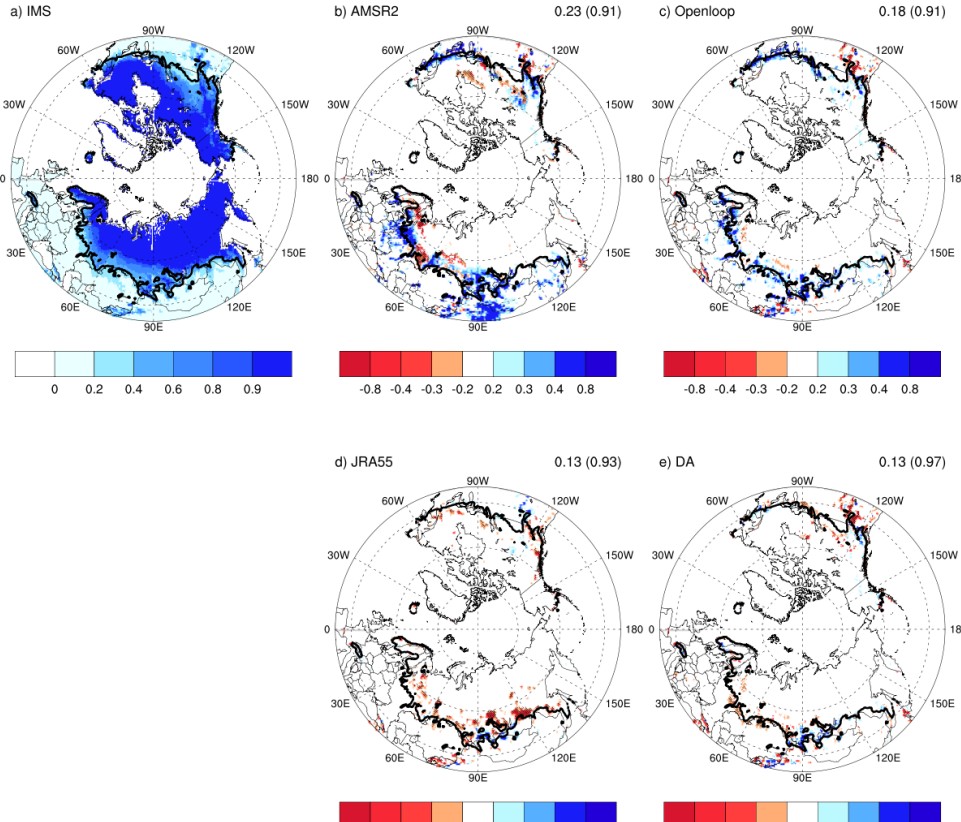


**Figure 2.** (a) Climatology of SCF from IMS used as reference and (b-e) the differences from
IMS for AMSR2, base-line model simulation (Openloop), JRA55, and the data
assimilation results (DA) for April during 2013-2020. The black line represents the
boundary of the transition region, defined as the climatological-mean SWE of less than
16mm. Each value on the top right is the root-mean-squared difference with IMS and
the accuracy from IMS (parenthesis) for 15323 pixels over 40-60°N. The accuracy is
defined in supplementary Table 1 as in previous study (Lee et al., 2015). Negative
values are indicated with a diagonal line.

780

781



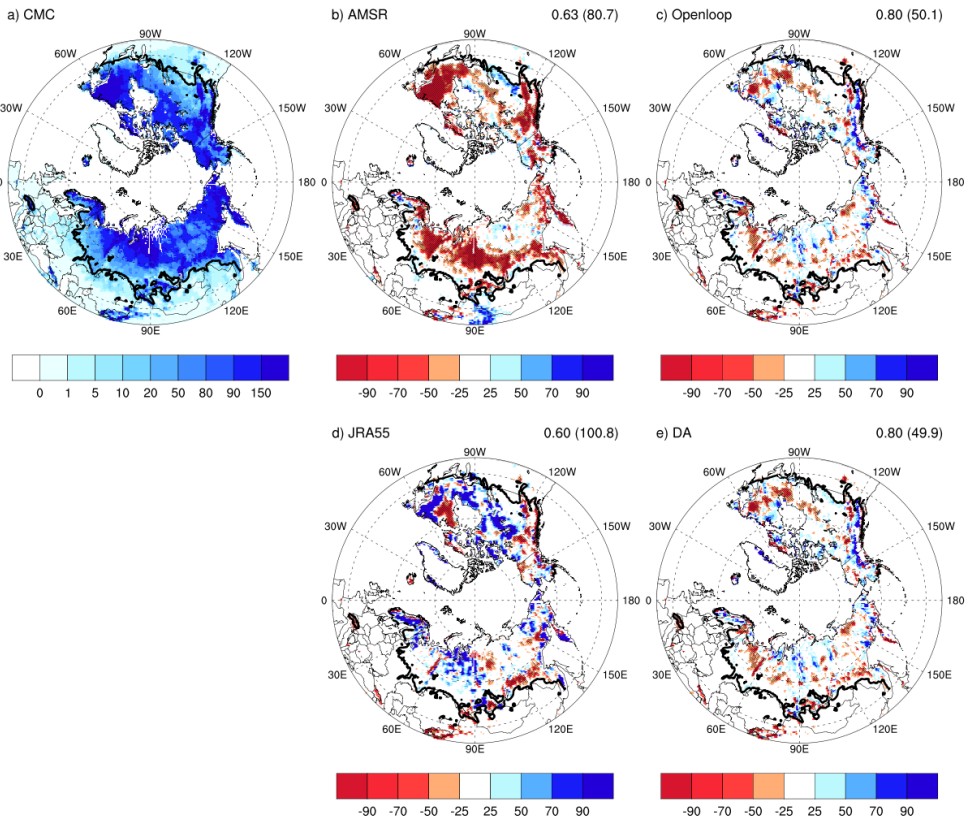

**Figure 3.** (a) Climatology of SWE from CMC used as reference and (b-e) the differences from CMC for AMSR2, base-line model simulation (Openloop), JRA55, and the data assimilation results (DA) for April during 2013-2020. The black line represents the boundary of the transition region, defined as the climatological-mean SWE of less than 16mm. Each value on the top right is the pattern correlation with CMC for 26482 pixels over 40 $^{\circ}$N and the root-mean-squared difference (unit: kg/m$^2$) from IMS (parenthesis) for 15323 pixels over 40-60$^{\circ}$N. Negative values are indicated with a diagonal line.

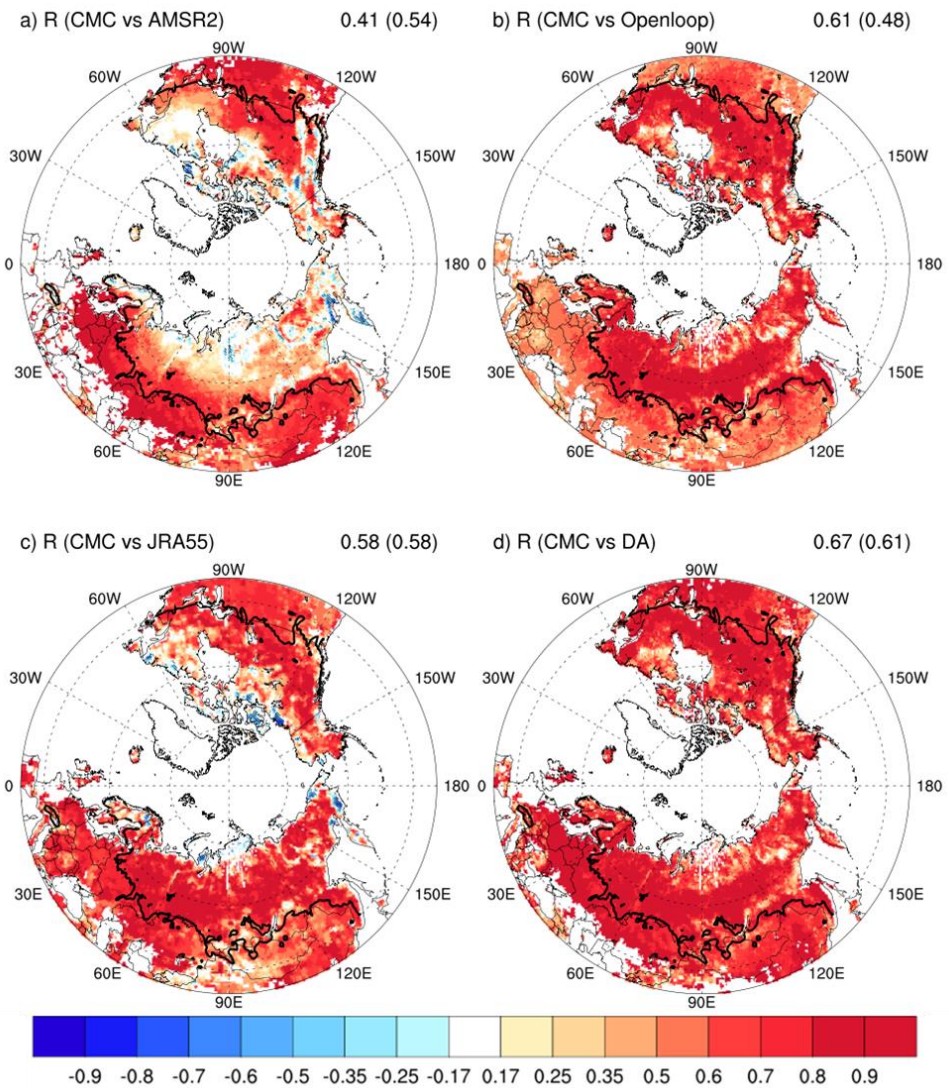

**Figure 4.** SWE skill measured as the Spearman rank correlation (R) with the CMC for AMSR2, base-line model simulation (Openloop), JRA55, and the data assimilation result (DA). The black line represents the boundary of the transition region, defined as the climatological-mean SWE of less than 16mm. Each value on the top is the area-average of North hemisphere for 26482 pixels over 40ºN and for 8801 pixels over the transition region (parenthesis). Negative values are indicated with a diagonal line.



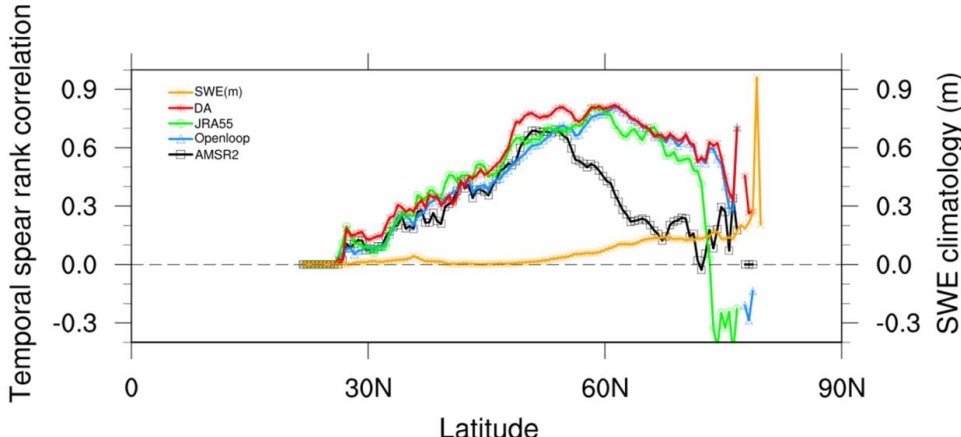

**Figure 5**. Zonally-averaged Spearman rank correlation (R) along the latitude for SWE. The yellow line indicates the climatology of SWE, and the black, blue, green, and red lines denote the values of AMSR2, base-line model simulation (Openloop), JRA55, data assimilation results (DA), respectively.



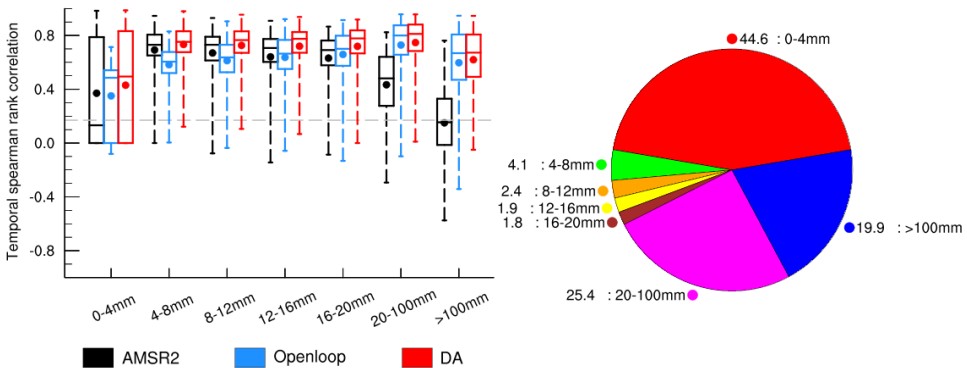

805

**Figure 6**. Box plots of the Spearman rank correlation (R) according to SWE. The pie chart

shows the total area ratio (%) as a function of SWE amount. The black, blue, and red

boxes denote the AMSR2, base-line model simulation (Openloop), and the data

assimilation results (DA), respectively. The boxes indicate 25 and 75% percentiles, and

the line and point in the boxes shows the median and the mean values. The upper and

lower whiskers denote the 10 and 90% percentiles, respectively.

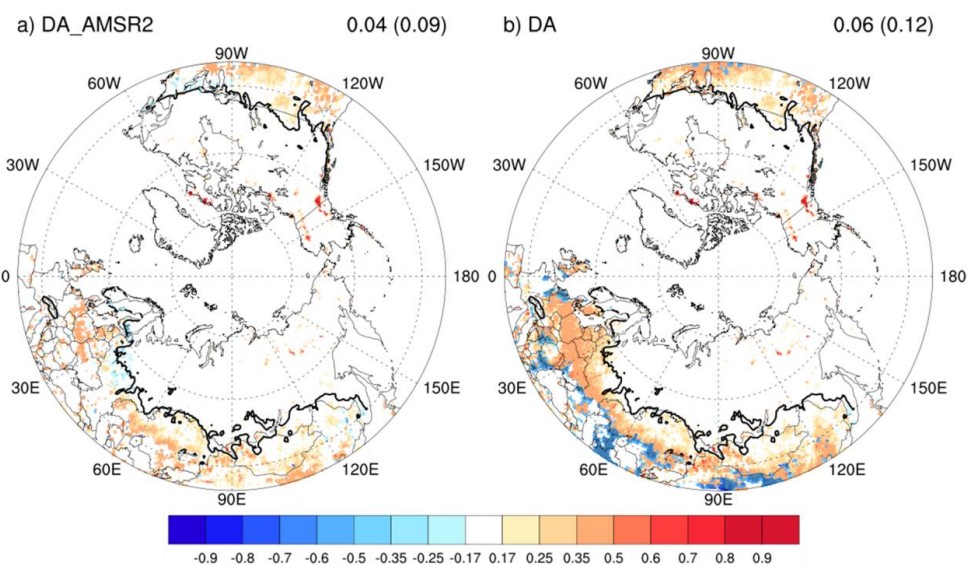

**Figure 7.** Difference in SWE skill, measured as the Spearman rank correlation coefficient (R) with CMC, between the Openloop and the data assimilation result employing both AMSR2 and IMS (referred to as DA), as well as the data assimilation result utilizing solely AMSR2 and excluding IMS (referred to as DA_AMSR2), for April during 2013-2020. The black line represents the boundary of the transition region, defined as the climatological-mean SWE of less than 16mm. Each value on the top right is the area-average over 40°N and the transition region (parenthesis). Negative values are indicated with a diagonal line.





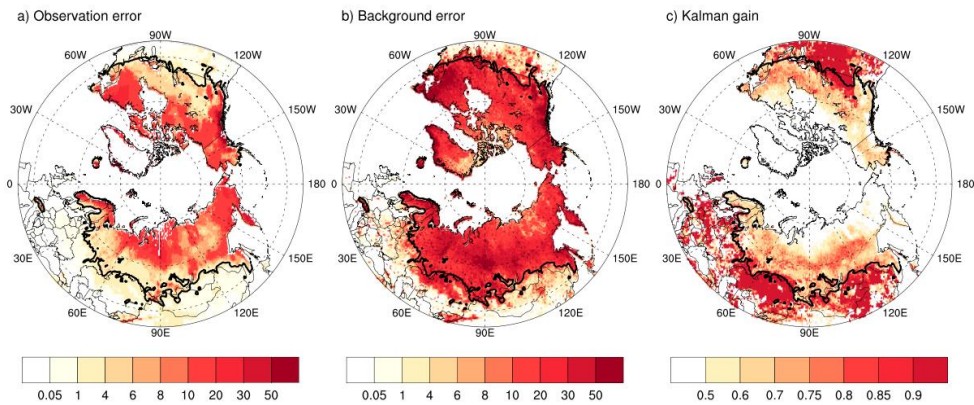

**Figure 8**. Spatial distribution of observation error, background error, and Kalman gain. The black line represents the boundary of the transition region, defined as the climatological-mean SWE of less than 16mm.



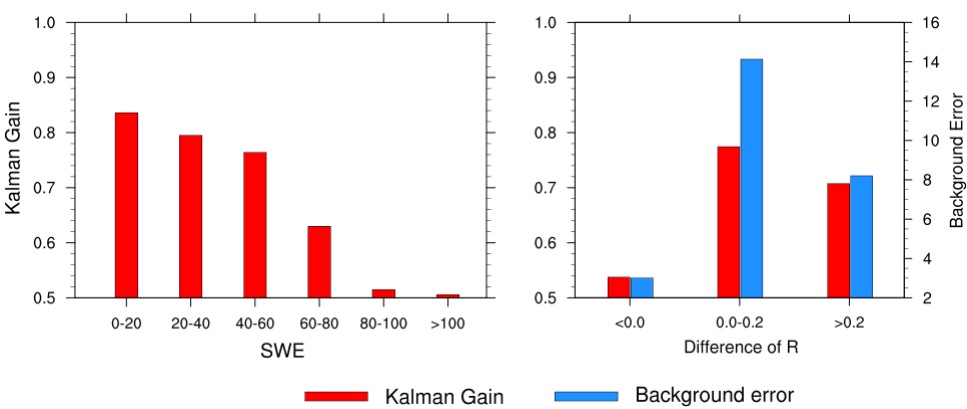

830

**Figure 9**. Bar chart of (left) the Kalman gain according to the SWE amount, and (right) the
Kalman gain (red line) and background error (blue line) according to the difference of
the Spearman rank correlation (R) between Openloop and DA.

834

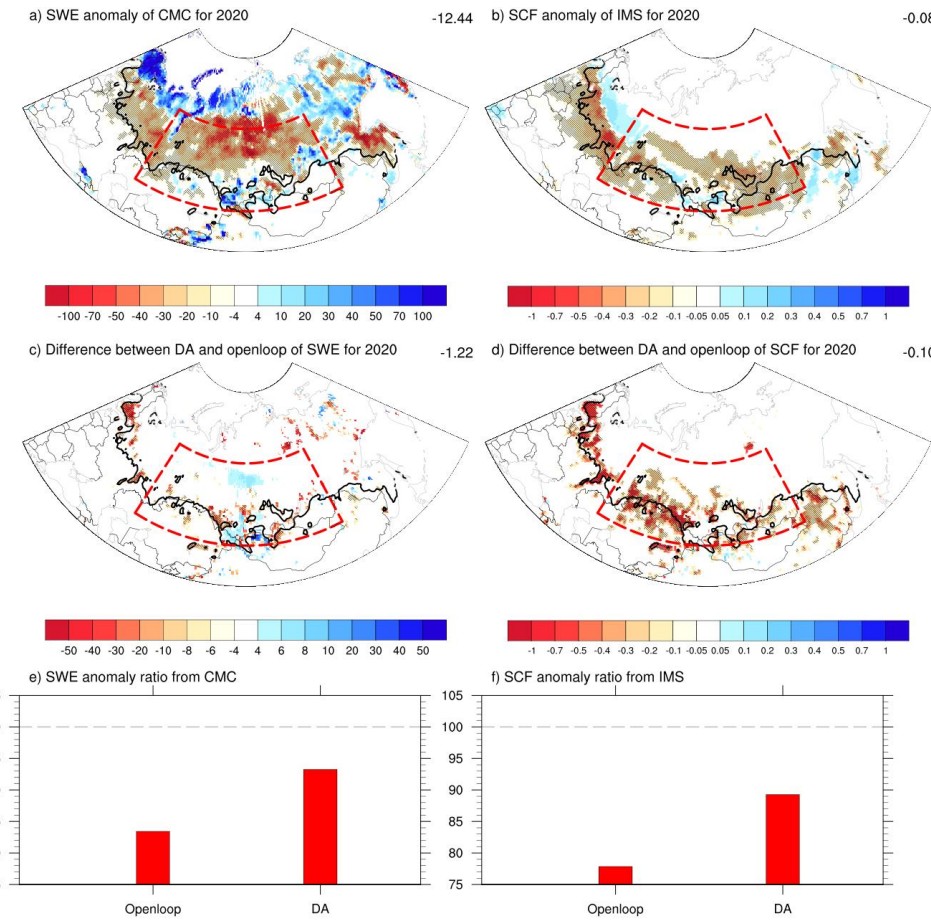

**Figure 10**. Anomalies of a) SWE from CMC and b) SCF from IMS as well as the difference (c, d) of variables between DA and openloop in April 2020. Bar chart (e, f) indicates the ratio of DA and openloop to verification data such as CMC and IMS in the red box (48–65ºN and 55–120ºE), which is the region associated with extreme high-temperature events, focused on this study. Negative values are indicated with a diagonal line.