# Peer review of "Assimilation of snow water equivalent from AMSR2 and IMS satellite data utilizing the local ensemble transform Kalman filter"

_Geoscientific Model Development, 2023_

## Referee Comment (RC2)

[referee-annotated manuscript omitted]

---

## Referee Comment (RC3)

Review of "Assimilation of snow water equivalent from AMSR2 and IMS satellite data utilizing the local ensemble transform Kalman filter"

**Summary**

This work assimilates the snow retrieval from AMSR2 (and the snow cover from IMS, albeit indirectly) into the JULES model using LETKF. The data assimilation (DA) framework offers an objective way to optimally combine the two imperfect dataset: the JULES model which has larger uncertainty in the transitional region, and the satellite retrieval which on the other hand exhibits greater uncertainty in the deep snow region. It is shown that the DA simulation is able to provide better initial conditions and forecast for snow, compared to the one without DA and other existing methods.

Overall, the DA approach and the experiment setup are carefully designed, the analyses are done well, and the results hold promise. However, there are concerns about the coherence in the current manuscript, especially in the introduction, making it difficult for the readers to follow and to understand the significance of this work. Therefore, I suggest a major revision in this iteration.

**Comments**

L70-117: Different snow states products derived using in-situ observation, remote-sensing retrievals, and using numerical models are summarized in these three paragraphs. However, these paragraphs appear disconnected. The coherence could be improved by trimming some unnecessary details, and emphasizing more on, e.g, (1) advantage/limitations of each dataset (2) the exactly snow state (i.e., SWE, SD, SCF, etc) that each dataset provides.

Following these paragraphs, e.g., a comparison/summary paragraph for these dataset could be presented, which can lead to the explanation why data assimilation or other data fusion methods are considered necessary/beneficial for constructing snow states.

L103-117: These methods (e.g., optimal interpolation) are similar to using data assimilation in the sense that they both combine the model simulation with the observations. You might want to emphasize why your DA system is a better method compared to these existing methods.

L135: The connection between this and the previous paragraph is unclear.

L143-146, 149-151: You may want to emphasize the unique contribution of this work compared to previous studies mentioned in these lines.

L162-163: "model error" -> I recommend change to "background error" (also in other places). Also, the Kalman gain measures the ratio of the background error to the sum of the background and the observation error.

L218-220: it's unclear how SCF is used based on the statement. You do explain it later in the text, but I suggest add something like 'this will be detailed later in Section …'

L251: I suggest change "due to randomness" -> "to account for the uncertainties in these variables"

L265-267: This sentence is unclear. The bias/mean of what?

L293: 'true probability distribution' -> 'flow-dependent probability distribution'

L294: 'LETKF applies an adaptive inflation scheme' -> 'LETKF is able to adopt an adaptive inflation scheme'.

L294-296: In most adaptive inflation schemes (for adjusting the ensemble spread), they are used to address to issues of insufficient ensemble spread, which mainly comes from the insufficient ensemble size (i.e., sampling errors) and model errors that are not properly accounted for. I suggest rephrase the sentence and delete the observational error.

In addition, since the adaptive inflation scheme (to adjust the ensemble spread) is not used in this work, maybe you could just remove it as it doesn't add much here.

L296-320: This paragraph needs to be rewritten. There are many details in the equations that are not explained. Since these equations are quite standard for LETKF, I would recommend trim down some details, and use plain language to briefly explain what LETKF is and how it works.

Also, you might want to introduce and define Kalman gain here as it is discussed in Section 4.2.

L317-318: You mentioned the adaptive inflation scheme, but here you apply a fixed inflation scheme. Have you tried using any adaptive inflation scheme to adjust the ensemble spread?

L321: You may want to add a few sentences to briefly explain what the localization is (and also why) here, before introducing the weight function.

L342-345: The assignment of the observation error seems to be a little arbitrary here. Are there any studies trying to estimate the observation error (e.g., using Desroziers et al. 2005) for this retrieval? I suggest elaborate more on the observation error as it is an important part of the DA system.

Desroziers, G., Berre, L., Chapnik, B. and Poli, P. (2005), Diagnosis of observation, background and analysis-error statistics in observation space. Q.J.R. Meteorol. Soc., 131: 3385-3396.

Minor comments for the DA setup:
(1) LETKF is optimal when the background error is Gaussian distributed. I suspect that in the transitional region, the ensemble distribution of SWE might not be Gaussian (e.g., when some ensemble members have snow while others do not). It might be interesting to have a look at the background and analysis ensemble at these grids.
(2) The observation error standard deviation is assigned to be proportional to the observed value. With this situation-dependent observation error, it is easier for DA to decrease the model SWE (when model > obs) as opposed to increase SWE (when model < obs). I am curious if this leads to negative biases in the SWE?

L433-434 (and Figure 7): Although overall DA is better than DA_AMSR2, there are more blue patches in DA compared to DA_AMSR2. DA_AMSR2 shows improvement almost globally. Do you have any comments on this?

L455 (and S Figure 1): It's hard to tell from (c) that the ratios are one especially in the transitional region. Nevertheless, the overall pattern in (a) and (b) looks similar, which is a good sign, suggesting the ensemble system does a decent job in quantifying the uncertainty. I suggest change (b) to 'ensemble spread'.

L478-481: I am not sure if I understand the causal relation implied here. Please clarify.

L486-489: To my understanding, the change in the land component does not feedback

to the atmospheric conditions in your DA setup (e.g., Figure 1). Therefore, I suspect if we can see the mechanism mentioned in L486-489 in your DA experiment. Do you really see the mechanism by comparing DA and OpenLoop, or is L486-489 simply an inference for a hypothetical scenario when a two-way coupled system is used? Please clarify.

L766 (Figure 1): "OBSERVAIONS" -> "OBSERVATIONS"

---

## Author Response (AR1)

Response to Reviews of ″Assimilation of snow water equivalent from AMSR2 and IMS satellite data utilizing the local ensemble transform Kalman filter″ by Joonlee lee, Myong-In Lee, Sunlae Tak, Eunkyo Seo, and Yong-Keun Lee. (Geoscientific Model Development: #gmd-2023-221)

We would like to thank the reviewers for their valuable feedback. Their insightful comments helped improve the quality of this paper. After examining the reviewers' comments, we have corrected and modified our manuscript. Our responses to the individual comments are provided below in blue.

**Reply to the Reviewer (#1)′s Comments:**

Snow accumulation can influence global or local energy balance by controlling surface reflectivity, and information related to snow water equivalent (SWE) plays a crucial role in local hydrological modeling and water resources management. With the rapid changes in global climate, snow information (coverage area, volume, reflectivity, and SWE) also exhibits varying degrees of fluctuations. In this context, Lee et al. developed a data assimilation method based on SWE data from AMSR2 and snow cover fraction data from IMS. The topic chosen for this manuscript holds certain scientific value and contributes to the current fields of snow remote sensing and cryosphere remote sensing. However, as an academic paper, the manuscript has serious issues in terms of writing, failing to convey the core content of the research. Specifically:

(1) Abstract:

The research background or scientific problems are not explained. The presentation of research results is unclear, and the conclusions are not specific. The manuscript fails to indicate its position and role in the field of snow remote sensing.

Response) Thank you for your comment. In response to the reviewer's feedback, we have revised the abstract to provide the background more clearly, articulate conclusions, and highlight contributions. It has been revised as follows:

Revision) (L26-L55) Snow Water Equivalent (SWE), as one of the land initial conditions, plays a crucial role in global or regional energy and water balance, thereby exerting a considerable impact on seasonal and sub-seasonal scale predictions owing to its enduring memory over 1 to

2 months. Despite its importance, most SWE initialization remains challenging due to its reliance on simple approaches based on spatially constrained observation. Therefore, this study developed the advanced SWE data assimilation framework with satellite remote-sensing data utilizing the local ensemble transform Kalman filter (LETKF) and the Joint U.K. Land Environment Simulator (JULES) land model. This constitutes a novel approach that has not been previously attempted, as it offers an objective way to optimally combine two imperfect data sources: the satellite SWE retrieval from the Advanced Microwave Scanning Radiometer 2 (AMSR2) and dynamically balanced SWE from JULES land model. In this framework, an algorithm is additionally considered to determine the assimilation process based on the presence or absence of snow cover from the Interactive Multisensor Snow and Ice Mapping System (IMS) satellite, renowned for its superior reliability.

The baseline model simulation from JULES without satellite data assimilation shows superior performance in high-latitude regions with heavy snow accumulation but relatively inferior in the transition regions with less snow and high spatial and temporal variation. Contrastingly, the AMSR2 satellite data exhibit better performance in the transition regions but poorer in the high latitudes, presumably due to the limitation of the satellite data in the penetrating depth. The data assimilation (DA) demonstrates the positive impacts by reducing uncertainty in the JULES model simulations in most areas, particularly in the mid-latitude transition regions. In the transition regions, the model background errors from the ensemble runs are significantly larger than the observation errors, emphasizing great uncertainty in the model simulations. The results of this study highlight the beneficial impact of data assimilation by effectively combining both land surface model and satellite-derived data according to their relative uncertainty, thereby controlling not only transitional regions but also satellite-constrained areas experiencing heavy snow accumulation. This assimilation framework is anticipated to contribute to a more precise prediction of atmospheric conditions by realistically capturing the interaction between the atmosphere and land, given the substantial influence of SWE on energy and water balance at the interface of the atmosphere and land.

**(2) Introduction:**
**The writing logic is poor. From the title of the manuscript, the author's research subject is snow water equivalent (SWE). However, the introduction does not clarify the main focus of the manuscript. Most of the content (L49-L134) elaborates on snow information,**

**and these details are insufficient to emphasize the significance of the author's research on SWE data assimilation. The research objectives are not concise (L152-L170).**

Response) In accordance with the reviewer's feedback, we have emphasized the significance of research on Snow Water Equivalent (SWE) data assimilation and streamlined the research objectives for clarity.

Revision) (L72-L84) In the subseasonal to seasonal (S2S) timescales, land initial states are crucial components in the S2S timescale predictions due to the inherent memory that changes slowly for 1 to 2 months in the climate system (e.g., Derome et al. 2005; Chen et al., 2010; Seo et al., 2019). In particular, the realistic snow initial states contribute to improving S2S prediction skills, as proven in several modeling studies. For example, previous studies (Orsolini et al., 2013; Jeong et al., 2013) demonstrated a considerable enhancement in prediction skill of 2m air temperature up to a lead time of 1-2 months across certain regions of Eurasia and the Arctic during winter, depending on snow initialization. Moreover, other studies (Orsolini et al., 2016; Li et al., 2019) have revealed that wave activity propagating toward the stratosphere, influenced by snow initial conditions in climate models, can induce changes in the polar vortex and contribute to the persistence of the North Atlantic Oscillation (NAO) and the AO. This emphasizes the significance of snow initialization in climate models as an essential process for enhancing prediction performance at the S2S timescales.

For determining initial snow states, snow water equivalent (SWE) is explicitly used in most prediction models as a prognostic variable to constrain water and energy conservation (Li et al., 2019; Gan et al., 2021). SWE is generally provided from in-situ observation data, remote-sensing retrievals from satellites, or numerical models such as the stand-alone land surface models (LSMs) forced by observed atmospheric variables.

(L123-L125) Given that the majority of the reanalysis datasets rely on snow depth measurements, the SWE estimation is likely to introduce potential accuracy concerns when the snow depth information is combined with the sow density calculations.

(L134-L145) However, in regions where ground observations are unavailable, large errors may exist in the snow model outputs due to uncertainties in atmospheric forcing and imperfect model parameterization (Boone et al., 2004; Essery et al., 2009). Often, the snow processes parameterized in LSMs rely on observed properties sampled in limited areas (Lim et al., 2022).

In addition, as IMS snow cover only identifies the presence of snow, the data assimilation with the satellite snow cover only is not sufficient and inappropriate in constraining water and energy conservation. Alternative methods that consider the physical quantity of snow are required for the snow initialization.

One approach to mitigate the spatial discontinuity of ground observations is to use satellite-derived SWE with wide spatial coverage and frequent temporal resolution. However, the SWE retrievals from satellites still have considerable uncertainties.

(L154-L178) However, most previous studies have focused on targeted regions with limited ground-based observations. Snow initialization in global coverage using satellite-derived SWE remains a persistently challenging task.

Therefore, this study developed an advanced SWE data assimilation framework with satellite remote-sensing data using the local ensemble transform Kalman filter (LETKF) and the Joint U.K. Land Environment Simulator (JULES) land model. This constitutes a novel approach that has not been previously attempted, and it offers an objective way to optimally combine two imperfect data sources: the satellite SWE from the Advanced Microwave Scanning Radiometer 2 (AMSR2) and the dynamically-balanced SWE from the JULES land model forced by observed atmospheric fields. The estimated SWE data exhibit better consistence by additionally using snow cover data from the IMS data. This assimilation framework also enables the assessment of improvement as it provides insights into the reasons behind the performance improvement based on the Kalman gain analysis that measures the relative significance of the input data between the satellite and the land model during the data assimilation cycle. The satellite data have demonstrated high reliability in the transition regions of climatologically-shallow snow conditions (Gan et al., 2021), and these regions are known as "hot spots" of strong atmosphere-land coupling through snow melting and associated surface energy and water balance changes (Koster et al., 2004; Dirmeyer, 2011; Huning and AghaKouchak, 2020). From these perspectives, it would be important to evaluate the impact of satellites on the transition regions as well as on the deep accumulation regions where accurate satellite retrievals are challenging. Furthermore, the benefits of assimilating satellite retrievals in extremely high-temperature events, such as the case in April 2020 over Eurasia, can be elucidated. In this regard, we expect that this snow data assimilation framework with satellite-derived SWE can be significant in providing optimal snow initial states for improving the S2S prediction by global climate models.

**(3) Conclusion:**

**Repetitive and verbose, the conclusion section should succinctly state the model's strengths, shortcomings, and prospects, expressing the important role of this model in snow remote sensing. It is recommended to divide it into discussion and conclusion sections.**

Response) Following the reviewer's suggestions, we have eliminated redundant sentences in the conclusion part and instead emphasized the contributions of this study to the field of remote sensing. I have revised the final session to "Conclusion and Discussion" and added a discussion section after the conclusion.

[revised manuscript text omitted]

**In conclusion, the SWE data assimilation method proposed by the author has a certain promoting effect on the field of snow remote sensing (maybe). However, the manuscript's writing is poor, and a significant revision is suggested before resubmission to provide readers with a clear and concise manuscript.**

Response) Thank you for your insightful feedback, which has been instrumental in improving the quality of our manuscript. In response to the reviewer's comments, we have diligently

revised the manuscript to ensure clarity and conciseness, thereby facilitating better understanding for the readers.

**Reply to the Reviewer (#2)′s Comments :**

**While this paper is conveying some important information, it is two main drawbacks: (1) writing is sloppy and (2) the novel contributions of this study are unclear.**

Response) Thank you for your comment. We have thoroughly revised the manuscript to address the reviewer's comments, focusing on enhancing clarity and conciseness while incorporating the new contributions of this study as suggested. We believe these revisions have significantly improved the manuscript and are grateful for the valuable feedback provided.

**Data assimilation method used in this study is clearly not novel component of the paper, as Kalman filter has been used in many other studies. In my view, the authors can focus on DA of AMSR2 and JULES model, as the novel component. Are there any other studies assimilating these two datasets? If not, then this can be a good contribution of this paper. Otherwise, I ask authors to detail what is it they have contributed through this which not known already. In any case, a significant revision of introduction is required to put this study into the appropriate context.**

Response) As suggested by the reviewer, we have revised the introduction to emphasize that the method of combining the incomplete datasets of SWE from AMSR2 and SWE from the JULES model through data assimilation represents a novel approach that has not been previously attempted.

Revision) (L154-L179) However, most previous studies have focused on targeted regions with limited ground-based observations. Snow initialization in global coverage using satellite-derived SWE remains a persistently challenging task.

Therefore, this study developed an advanced SWE data assimilation framework with satellite remote-sensing data using the local ensemble transform Kalman filter (LETKF) and the Joint U.K. Land Environment Simulator (JULES) land model. This constitutes a novel approach that has not been previously attempted, and it offers an objective way to optimally combine two imperfect data sources: the satellite SWE from the Advanced Microwave Scanning Radiometer 2 (AMSR2) and the dynamically-balanced SWE from the JULES land model forced by

observed atmospheric fields. The estimated SWE data exhibit better consistence by additionally using snow cover data from the IMS data. This assimilation framework also enables the assessment of improvement as it provides insights into the reasons behind the performance improvement based on the Kalman gain analysis that measures the relative significance of the input data between the satellite and the land model during the data assimilation cycle. The satellite data have demonstrated high reliability in the transition regions of climatologically-shallow snow conditions (Gan et al., 2021), and these regions are known as "hot spots" of strong atmosphere-land coupling through snow melting and associated surface energy and water balance changes (Koster et al., 2004; Dirmeyer, 2011; Huning and AghaKouchak, 2020). From these perspectives, it would be important to evaluate the impact of satellites on the transition regions as well as on the deep accumulation regions where accurate satellite retrievals are challenging. Furthermore, the benefits of assimilating satellite retrievals in extremely high-temperature events, such as the case in April 2020 over Eurasia, can be elucidated. In this regard, we expect that this snow data assimilation framework with satellite-derived SWE can be significant in providing optimal snow initial states for improving the S2S prediction by global climate models.

**I have made several comments on the attached pdf which will help the authors improve the writing but there are many more sentence that may will benefit from a revision.**
Response) We have thoroughly reviewed the attached PDF containing the reviewer's comments and made every effort to incorporate them into the manuscript.

**An important point, I was perpetually confused about for what quantity are the author trying to use data assimilation? SWE, SD or SCF? It should be clearly mentioned in the introduction.**
Response) We have revised the introduction to clearly mention the Snow Water Equivalent (SWE) data assimilation of our study.

Revision) (L72-L88) In the subseasonal to seasonal (S2S) timescales, land initial states are crucial components in the S2S timescale predictions due to the inherent memory that changes slowly for 1 to 2 months in the climate system (e.g., Derome et al. 2005; Chen et al., 2010; Seo et al., 2019). In particular, the realistic snow initial states contribute to improving S2S

prediction skills, as proven in several modeling studies. For example, previous studies (Orsolini et al., 2013; Jeong et al., 2013) demonstrated a considerable enhancement in prediction skill of 2m air temperature up to a lead time of 1-2 months across certain regions of Eurasia and the Arctic during winter, depending on snow initialization. Moreover, other studies (Orsolini et al., 2016; Li et al., 2019) have revealed that wave activity propagating toward the stratosphere, influenced by snow initial conditions in climate models, can induce changes in the polar vortex and contribute to the persistence of the North Atlantic Oscillation (NAO) and the AO. This emphasizes the significance of snow initialization in climate models as an essential process for enhancing prediction performance at the S2S timescales.

Snow states, i.e., snow water equivalent (SWE) used directly for hydrological analysis and initial states of the model (Li et al., 2019; Gan et al., 2021), are generally provided from in-situ observations data, remote-sensing retrievals from satellites, or numerical models such as the land surface model (LSM) operated based on the observed atmospheric variables.

**On the methodological side, the method is well implemented but the description of the LETKF is quite unclear. I have a basic understanding of LETKF, but I could not understand what authors are trying to explain in section 3.2. Also, there needs to be some more discussion about JRA55 in the 'Data' section. What purpose does this data serve? In the results also, JRA55 is not discussed much.**

Response) In the data assimilation methods section, we have provided a concise explanation of the concepts, as we utilized the standard LETKF method. Additionally, we have employed JRA55 as the reanalysis dataset used for meteorological forcings in the JULES model, facilitating comparison with other reanalysis datasets in the context of our study results. Following the reviewer's suggestion, we have also included a discussion on JRA55 in the results section.

[revised manuscript text omitted]

**Overall, the paper needs a major revision before it can be judged for its significance to scientific literature. I have made several comments in the attached PDF which I hope help the authors in improving their work.**

Response) Thank you for your valuable input, which has greatly enhanced the clarity and impact of our paper. The PDF containing the reviewer's comments has been immensely helpful in improving the manuscript. We have extensively revised the text to incorporate the reviewer's feedback, thereby highlighting the scientific contributions and significance of our study.⌗

**Reply to the Reviewer (#3)′s Comments :**

**Review of "Assimilation of snow water equivalent from AMSR2 and IMS satellite data utilizing the local ensemble transform Kalman filter"**

**Summary**

**This work assimilates the snow retrieval from AMSR2 (and the snow cover from IMS, albeit indirectly) into the JULES model using LETKF. The data assimilation (DA) framework offers an objective way to optimally combine the two imperfect dataset: the JULES model which has larger uncertainty in the transitional region, and the satellite retrieval which on the other hand exhibits greater uncertainty in the deep snow region. It is shown that the DA simulation is able to provide better initial conditions and forecast for snow, compared to the one without DA and other existing methods.**

**Overall, the DA approach and the experiment setup are carefully designed, the analyses are done well, and the results hold promise. However, there are concerns about the coherence in the current manuscript, especially in the introduction, making it difficult for the readers to follow and to understand the significance of this work. Therefore, I suggest a major revision in this iteration.**

**Comments**

**L70-117: Different snow states products derived using in-situ observation, remote-sensing retrievals, and using numerical models are summarized in these three paragraphs. However, these paragraphs appear disconnected. The coherence could be improved by trimming some unnecessary details, and emphasizing more on, e.g, (1) advantage/limitations of each dataset (2) the exactly snow state (i.e., SWE, SD, SCF, etc) that each dataset provides. Following these paragraphs, e.g., a comparison/summary paragraph for these dataset could be presented, which can lead to the explanation why data assimilation or other data fusion methods are considered necessary/beneficial for constructing snow states.**

Response) Thank you for your comment. As suggested by the reviewer′s comment, we have revised these paragraphs related to different snow states products derived using in-situ observation, remote-sensing retrievals, and using numerical models

Revision) (L85-L125) Snow states, i.e., snow water equivalent (SWE) used directly for hydrological analysis and initial states of the model (Li et al., 2019; Gan et al., 2021), are generally provided from in-situ observations data, remote-sensing retrievals from satellites, or numerical models such as the land surface model (LSM) operated based on the observed atmospheric variables. For the in-situ data snow depth (SD) measurements prevail, largely attributed to the challenges associated with acquiring precise SWE data (Takala et al., 2011; De Rosnay et al., 2014). Surface synoptic observations (SYNOP) serve as the principal source for SD measurements. The in-situ measurements offer the most dependable snow information, yet they are characterized by relatively coarse temporal and spatial resolutions, particularly within limited areas, due to the spatial heterogeneity inherent in snow distribution. (Helmert et al., 2018; Meyal et al., 2020). Satellite-derived observations using conical scanning microwave instruments may provide spatially consistent data coverage across the globe. Cho et al. (2017) showed the SWE retrieval results from two passive microwave sensors, the advanced microwave scanning radiometer 2 (AMSR2) and the special sensor microwave imager sounder (SSMIS). However, the algorithms for SWE retrieval exhibit a degree of sensitivity to a variety of parameters such as snow liquid water content and snow grain size distribution (De Rosnay et al., 2014). Hence, satellite-based SWE data still have limitations in accuracy, especially under deep snow conditions due to the limited penetration depth (Gan et al., 2021). On the other hand, satellite retrieval can estimate snow cover accurately under clear sky conditions (Brubaker et al., 2009). Model simulations obtained from LSMs and simple snow models can cover complete spatiotemporal resolution but involve potentially large uncertainties due to the deficiencies in the physical parameterizations and meteorological forcing data (Dirmeyer et al., 2006; Seo et al., 2021).

Considering that snow observation datasets have their respective strengths as well as limitations, data assimilation or other data fusion methods can prove to be beneficial for constructing snow states such as reanalysis data (e.g., Brasnett, 1999; Dee et al., 2011; Meng et al., 2012; Pullen et al., 2011; De Rosnay et al., 2014). For example, the snow analysis for the Canadian Meteorological Center (CMC) utilizes a 2-dimensional optimal interpolation (2D-OI) scheme with in-situ observations and the outputs from a simple snow model (Brown et al., 2003). The National Centers for Environmental Prediction (NCEP) climate forecast system reanalysis (CFSR) combines a multi-satellite-based interactive multi-sensor snow and ice mapping system (IMS) as satellite-based snow cover retrieval and the outputs from the global snow model of the Air Force Weather Agency (Meng et al., 2012). At the European

Center for Medium Weather Forecast (ECMWF), the ECMWF reanalysis (ERA)-Interim and ERA5 for the snow analysis employ a Cressman interpolation and 2D-OI, respectively, with the IMS, in-situ observation, and the results from a land surface model (Dee et al. 2011; De Rosnay et al., 2014). The Japanese 55-year Reanalysis (JRA55) also utilizes the 2D-OI with in-situ observation, satellite-based snow cover from SSMIS, and the results from an LSM (Kobayashi et al., 2015). Given that the majority of the reanalysis datasets rely on snow depth measurements, the SWE estimation is likely to introduce potential accuracy concerns when the snow depth information is combined with the sow density calculations.

**L103-117: These methods (e.g., optimal interpolation) are similar to using data assimilation in the sense that they both combine the model simulation with the observations. You might want to emphasize why your DA system is a better method compared to these existing methods.**

Response) We have revised the manuscript to highlight the reasons why our data assimilation system outperforms existing methods.

[revised manuscript text omitted]

**L143-146, 149-151: You may want to emphasize the unique contribution of this work compared to previous studies mentioned in these lines.**

Response) We have revised the sentences to emphasize the unique contributions of our study compared to previous research.

Revision) (L154-L179) However, most previous studies have focused on targeted regions with limited ground-based observations. Snow initialization in global coverage using satellite-derived SWE remains a persistently challenging task.

Therefore, this study developed an advanced SWE data assimilation framework with satellite remote-sensing data using the local ensemble transform Kalman filter (LETKF) and the Joint U.K. Land Environment Simulator (JULES) land model. This constitutes a novel approach that has not been previously attempted, and it offers an objective way to optimally combine two

imperfect data sources: the satellite SWE from the Advanced Microwave Scanning Radiometer 2 (AMSR2) and the dynamically-balanced SWE from the JULES land model forced by observed atmospheric fields. The estimated SWE data exhibit better consistence by additionally using snow cover data from the IMS data. This assimilation framework also enables the assessment of improvement as it provides insights into the reasons behind the performance improvement based on the Kalman gain analysis that measures the relative significance of the input data between the satellite and the land model during the data assimilation cycle. The satellite data have demonstrated high reliability in the transition regions of climatologically-shallow snow conditions (Gan et al., 2021), and these regions are known as "hot spots" of strong atmosphere-land coupling through snow melting and associated surface energy and water balance changes (Koster et al., 2004; Dirmeyer, 2011; Huning and AghaKouchak, 2020). From these perspectives, it would be important to evaluate the impact of satellites on the transition regions as well as on the deep accumulation regions where accurate satellite retrievals are challenging. Furthermore, the benefits of assimilating satellite retrievals in extremely high-temperature events, such as the case in April 2020 over Eurasia, can be elucidated. In this regard, we expect that this snow data assimilation framework with satellite-derived SWE can be significant in providing optimal snow initial states for improving the S2S prediction by global climate models.

**L162-163: "model error" -> I recommend change to "background error" (also in other places). Also, the Kalman gain measures the ratio of the background error to the sum of the background and the observation error.**

Response) As the point well taken, we corrected it.

**L218-220: it's unclear how SCF is used based on the statement. You do explain it later in the text, but I suggest add something like 'this will be detailed later in Section …'**

Response) Thank you for your comment. We have added a sentence indicating that further explanations will be provided later in the manuscript, as per the reviewer's comment.

Revision) (L205-L207) In this study, the application of the assimilation process is determined based on IMS-based SCF, renowned for its superior reliability (e.g., Brown et al., 2014).

Further details will be described in Section 3.3.

**L251: I suggest change "due to randomness" -> "to account for the uncertainties in these variables"**

Response) As the point well taken, we corrected it.

**L265-267: This sentence is unclear. The bias/mean of what?**

Response) The sentence has been revised to provide a clear explanation, aiming to enhance understanding.

Revision) (L275-L278) The discrepancy in SWE between remote sensing and LSMs often arises due to uncertainties in the model physics and forcing data and satellite retrievals. These uncertainties can lead to a significant discrepancy in SWE between model simulations and satellite remote-sensing retrievals, potentially degrading performance.

**L293: 'true probability distribution' -> 'flow-dependent probability distribution'**

Response) As the point well taken, we corrected it.

**L294: 'LETKF applies an adaptive inflation scheme' -> 'LETKF is able to adopt an adaptive inflation scheme'.**

Response) As the point well taken, we corrected it.

Revision) (L315-L317) Third, the LETKF employs an inflation parameter to adjust the ensemble spread, ensuring realistic uncertainty estimation by accounting for background errors.

**L294-296: In most adaptive inflation schemes (for adjusting the ensemble spread), they are used to address to issues of insufficient ensemble spread, which mainly comes from the insufficient ensemble size (i.e., sampling errors) and model errors that are not**

**properly accounted for. I suggest rephrase the sentence and delete the observational error. In addition, since the adaptive inflation scheme (to adjust the ensemble spread) is not used in this work, maybe you could just remove it as it doesn't add much here.**

Response) The sentence has been revised in accordance with the reviewer's comment.

Revision) (L315-L321) Third, the LETKF employs an inflation parameter to adjust the ensemble spread, ensuring realistic uncertainty estimation by accounting for background errors. The underestimation of the analysis error covariance is typically issued by spatially and temporally constant boundary conditions and observation errors and limited ensemble members. Based on the standardized LETKF, this study applies a multiplicative covariance inflation of 20% of the spread of 24 member ensembles for each data assimilation cycle.

**L296-320: This paragraph needs to be rewritten. There are many details in the equations that are not explained. Since these equations are quite standard for LETKF, I would recommend trim down some details, and use plain language to briefly explain what LETKF is and how it works. Also, you might want to introduce and define Kalman gain here as it is discussed in Section 4.2.**

Response) Thank you for your comment. In response to the reviewer's comment, we have streamlined and provided a concise explanation of some details regarding LETKF. Additionally, we have included the definition of Kalman gain in this section.

Revision) (L299-L324) The snow assimilation is conducted based on the LETKF (e.g., Hunt et al., 2007), which is utilized to combine remotely sensed retrievals with the LSM model outputs (a.k.a. backgrounds) to produce a snow analysis. Unlike variational data assimilation methods, non-variational approaches (i.e., ensemble-based filters) characterize a probabilistic representation with the spread of the ensemble serving as an estimate of forecast uncertainty. LETKF has several advantages over other data assimilation methods. First, LETKF can efficiently handle large datasets and high-dimensional state variables by localizing the covariance matrix. This offers efficiency in parallel computing, making it suitable for real-time forecasting and high-resolution data assimilation. In this study, the horizontal local patch size and the localization length scale parameters are defined as 150 km and 30 km (Table 1), respectively. This approach involves the weight function for the covariance localization within

the local patch centered at the analysis grid (e.g., Houtekamer and Mitchell, 2001; Hamill et al., 2001). This function assigns larger errors to observations located farther away from the center of the local patch, as proposed by Miyoshi and Yamane (2007), depending on the Gaussian function. Secondly, the method utilizes model simulation ensembles to capture the uncertainty in the initial states and background errors, which allows for a better representation of the flow-dependent probability distribution of the state variables that vary in time and space. Third, the LETKF employs an inflation parameter to adjust the ensemble spread, ensuring realistic uncertainty estimation by accounting for background errors. The underestimation of the analysis error covariance is typically issued by spatially and temporally constant boundary conditions and observation errors and limited ensemble members. Based on the standardized LETKF, this study applies a multiplicative covariance inflation of 20% of the spread of 24 member ensembles for each data assimilation cycle. Furthermore, the Kalman gain analysis (Seo et al., 2021), which quantifies the ratio of the background error to the total error (equivalent to the sum of the background and the observation error), is conducted. This analysis serves to determine the weights assigned to assimilated observations in the analysis update processes of the LETKF.

**L317-318: You mentioned the adaptive inflation scheme, but here you apply a fixed inflation scheme. Have you tried using any adaptive inflation scheme to adjust the ensemble spread?**

Response) In this study, we employ an inflation parameter based on a previous study (e.g., Seo et al., 2021), instead of utilizing the adaptive inflation scheme. The ensemble spread in this study demonstrates a sufficiently valid magnitude in comparison with the RMSE, as illustrated in SFig. 1, indicating that it is well estimated.

Revision) (L315-L321) Third, the LETKF employs an inflation parameter to adjust the ensemble spread, ensuring realistic uncertainty estimation by accounting for background errors. The underestimation of the analysis error covariance is typically issued by spatially and temporally constant boundary conditions and observation errors and limited ensemble members. Based on the standardized LETKF, this study applies a multiplicative covariance inflation of 20% of the spread of 24 member ensembles for each data assimilation cycle.

**L321: You may want to add a few sentences to briefly explain what the localization is(and also why) here, before introducing the weight function.**

Response) Thank you for your comment. We have added a brief explanation about localization in accordance with the reviewer's comment.

Revision) (L304-L312) First, LETKF can efficiently handle large datasets and high-dimensional state variables by localizing the covariance matrix. This offers efficiency in parallel computing, making it suitable for real-time forecasting and high-resolution data assimilation. In this study, the horizontal local patch size and the localization length scale parameters are defined as 150 km and 30 km (Table 1), respectively. This approach involves the weight function for the covariance localization within the local patch centered at the analysis grid (e.g., Houtekamer and Mitchell, 2001; Hamill et al., 2001). This function assigns larger errors to observations located farther away from the center of the local patch, as proposed by Miyoshi and Yamane (2007), depending on the Gaussian function.

**L342-345: The assignment of the observation error seems to be a little arbitrary here. Are there any studies trying to estimate the observation error (e.g., using Desroziers et al. 2005) for this retrieval? I suggest elaborate more on the observation error as it is an important part of the DA system. Desroziers, G., Berre, L., Chapnik, B. and Poli, P. (2005), Diagnosis of observation, background and analysis-error statistics in observation space. Q.J.R. Meteorol. Soc., 131: 3385-3396.**

Response) As mentioned by reviewers, observational error in data assimilation is a crucial aspect within the context. Regrettably, there is a lack of previous studies addressing the accurate observational errors pertaining to AMSR2 SWE. Therefore, the observation error in this study is conservatively prescribed as 10% of AMSR2 SWE for each grid compared to the previous study utilizing AMSR2 SWE data (Lee et al., 2015), considering the general increase in the errors during the snow accumulation period with the development of deep snowpack. It has been revised as follows:

Revision) (L335-L339) Due to the absence of precise error estimates for AMSR2 SWE retrievals, the observation error is conservatively prescribed as 10% of AMSR2 SWE for each

grid compared to the previous study utilizing AMSR2 SWE data (Lee et al., 2015), considering the general increase in the errors during the snow accumulation period with the development of deep snowpack (Foster et al., 2005; Cho et al., 2017).

**Minor comments for the DA setup:**

**(1) LETKF is optimal when the background error is Gaussian distributed. I suspect that in the transitional region, the ensemble distribution of SWE might not be Gaussian (e.g., when some ensemble members have snow while others do not). It might be interesting to have a look at the background and analysis ensemble at these grids.**

Response) Thank you for your comment. Unlike soil moisture, SWE presents varying characteristics in the CDF distribution across different regions, such as between high and low latitudes or when some regions have snow while others do not, thus requiring the estimation of distribution at each grid point. However, the SWE distribution among ensemble members consistently exhibited a Gaussian distribution (Additional_Fig.1). Particularly, a distinct Gaussian distribution was evident in transitional regions, indicating its association with performance enhancement through data assimilation with optimized background error distribution. In response to this aspect, the following content has been incorporated into the manuscript.

[Figure]

Additional_Figure1 (Supplementary Fig. 4). Probability Density Function (PDF) of ensemble

distribution for Snow Water Equivalent (SWE) over Global (red line) and Transitional Region (red line) for April during 2013-2020.

Revision) (L460-L463) The ensemble spread in this study demonstrates a sufficiently valid magnitude in comparison with the RMSE, as illustrated in SFig. 1, indicating that it is well estimated. Moreover, the SWE distribution among ensemble members consistently exhibited a Gaussian distribution, with a distinct this distribution particularly evident in transitional regions (SFig. 4).

**(2) The observation error standard deviation is assigned to be proportional to the observed value. With this situation-dependent observation error, it is easier for DA to decrease the model SWE (when model > obs) as opposed to increase SWE (when model < obs). I am curious if this leads to negative biases in the SWE?**

Response) Thank you for your comment. In Figure 8, similar to observational errors (Fig. 8a), it is apparent that the background error exhibits considerable variation depending on the quantity of snow (Fig. 8b). Also, due to standard normal deviation scaling as bias correction applied to the satellite data utilized in data assimilation (Additional _Fig.2b), no discernible structural negative bias in the data assimilation results is evident (Additional _Fig.2d). Through additional figures, we can observe that the areas where bias is improved via data assimilation are predominantly transitional regions (Additional _Fig.2e).

[Figure]

Additional_Figure2. Mean bias of SWE from CMC for AMSR2 (a), bias-corrected AMSR (b), Openloop (c), and DA (d), and difference (e: d-c) for April during 2013-2020. The black line represents the boundary of the transition region, defined as the climatological-mean SWE of less than 16mm. Each value on the top right is the pattern correlation with CMC for 26482 pixels over 40N and the root-mean-squared difference (unit: kg/m2) from CMC (parenthesis) for 15323 pixels over 40-60N. Negative values in red shades are indicated with a diagonal line.

[Figure]

Figure 8. Spatial distribution of observation error (unit: kg/m2), background error (unit: kg/m2),

and Kalman gain. The black line represents the boundary of the transition region, defined as the climatological-mean SWE of less than 16mm.

**L433-434 (and Figure 7): Although overall DA is better than DA_AMSR2, there are more blue patches in DA compared to DA_AMSR2. DA_AMSR2 shows improvement almost globally. Do you have any comments on this?**

Response) DA exhibits inferior performance compared to Openloop in certain exceptional cases, which may be attributed to discrepancies in snow identification between the CMC observations used for correlation and the IMS data utilized for data assimilation. Based on the reviewer′s comment, we have modified the sentence as follows:

Revision) (L439-L442) Notably, the skill is enhanced significantly in DA by incorporating the IMS SCF. DA exhibits inferior performance compared to Openloop in certain exceptional cases, which may be attributed to discrepancies in snow identification between the CMC observations used for correlation and the IMS data utilized for data assimilation.

**L455 (and S Figure 1): It's hard to tell from (c) that the ratios are one especially in the transitional region. Nevertheless, the overall pattern in (a) and (b) looks similar, which is a good sign, suggesting the ensemble system does a decent job in quantifying the uncertainty. I suggest change (b) to 'ensemble spread'.**

Response) As the point well taken, we corrected it.

**L478-481: I am not sure if I understand the causal relation implied here. Please clarify.**

Response) We have clarified the implied causality in this sentence based on the reviewer's comment.

Revision) (L485-L488) Additionally, it has been revealed that the occurrence of high temperatures in the Siberian region is found to be closely associated with large-scale atmospheric waves in the upper atmosphere over the Eurasian region originating from the Atlantic (De Angelis et al., 2023).

**L486-489: To my understanding, the change in the land component does not feedbackto the atmospheric conditions in your DA setup (e.g., Figure 1). Therefore, I suspect if we can see the mechanism mentioned in L486-489 in your DA experiment. Do you really see the mechanism by comparing DA and OpenLoop, or is L486-489 simply an inference for a hypothetical scenario when a two-way coupled system is used? Please clarify.**

Response) Previous study (Collow et al., 2022) shows the land-atmosphere interaction utilizing a coupled atmosphere-land model. We have modified the potentially misleading sentence to more accurately represent the content of the paper as follows:

Revision) (L493-L497) Substantial snow melt can contribute to record-breaking heatwaves through albedo feedback and changes in the ratio of the latent and sensible heat fluxes from the exposed surface, coupled with favorable atmospheric circulation patterns (Collow et al., 2022). Collow et al. (2022) demonstrated that the exposed surface contributed to up to 20% of the temperature anomaly over Siberia in spring 2020.

**L766 (Figure 1): "OBSERVAIONS" -> "OBSERVATIONS"**

Response) As the point well taken, we corrected it.

---

## Author Response (AR2)

Response to Reviews of ″Assimilation of snow water equivalent from AMSR2 and IMS satellite data utilizing the local ensemble transform Kalman filter″ by Joonlee lee, Myong-In Lee, Sunlae Tak, Eunkyo Seo, and Yong-Keun Lee. (Geoscientific Model Development: #gmd-2023-221)

We would like to thank the reviewers for their valuable feedback. Their insightful comments helped improve the quality of this paper. After examining the reviewers' comments, we have corrected and modified our manuscript. Our responses to the individual comments are provided below in blue.

**Reply to the Reviewer (#1)′s Comments:**

I appreciate the authors' substantial effort to revise the manuscript and address the issues raised in the previous review. The manuscript has been improved, especially in the introduction, which now better highlights the value and the significance of the work. I have only a few very minor questions/suggestions, which I believe won't significantly affect the quality of the paper. Therefore, I consider the manuscript in good shape for publication.

L108: I suggest revise it as "Considering that each snow observation dataset has its respective strengths…"
Response) Corrected as suggested.

L342-345: This sentence is long. I suggest break it into several shorter sentences.
Response) Modified as suggested.

Revision) (L342-L345) In addition, the analysis state of this method is calculated based on the IMS snow cover fraction as follows (Fig. 1). If the SCF from IMS is zero, the snow analysis is set to 0; otherwise, it is derived through data assimilation.

L450: I suggest add "for SWE" after "the Kalman gain" to be more specific
Response) Modified as suggested.

L462-464: a few comments
(1) In SFig.4, does the x-axis refer to the ensemble perturbation of SWE (i.e., SWE of each

**ensemble member minus SWE ensemble mean), instead of the SWE? Can SWE be negative?**

Response) As the reviewer correctly pointed out, the SWE in SFig.4 represents the ensemble perturbations, with the ensemble mean removed from each ensemble member. To prevent any misunderstanding, we have revised the caption accordingly.

Revision) **Supplementary Fig. 4** Probability density function (PDF) of the standardized SWE ensemble perturbation, with the ensemble mean removed from each ensemble member, at each grid point. The PDF is averaged globally (blue line) and for the transition region (red line) for April from 2013 to 2020. The black line represents the standardized Gaussian function N(0,1).

**(2) It is not clear to me whether the distribution look like Gaussian at all.**

Response) We agree with the reviewer's comments. We have recalculated the standardized ensemble spread at each grid point and created a new figure accordingly. Consequently, we have revised the relevant text in the manuscript.

Revision) (L462-464) Moreover, the standardized distribution of SWE among the ensemble members exhibits a quasi-Gaussian distribution centered around zero, with the transition region showing a closer resemblance to a standardized Gaussian distribution (SFig. 4).

**(3) The Gaussian assumption in LETKF requires that, at each grid, the distribution of the uncertainty (in this study, this is sampled by 25 ensemble member at each grid point) is Gaussian. It seems to me SFig.4 is the distribution mixing all the ensemble states at all grid points(?!). If so, the distribution in SFig.4 will possibly not be Gaussian, but instead, a Gaussian-mixture. This is because the standard deviation of the ensemble distribution is probably different at different grid point.**

Response) Given that the standard deviation of the ensemble distribution varies at each grid point, we have recalculated the standardized ensemble spread for each grid point and created a new figure accordingly.

**(4) I don't insist doing the following analysis (as it won't add much to the paper), but here is one possible way to verify the Gaussian assumption.**
**-(a) Normalize the ensemble perturbation at each grid point respectively**

**-(b) Plot the histogram of all the normalized ensemble perturbation at all grid points (or the grid points within the transitional region)**

**-(c) Compare the histogram in (b) to a standard Gaussian N(0,1)**

Response) Based on the reviewer's suggestion, we have plotted the standardized PDF at each grid point instead of the histogram and added the standard Gaussian N(0,1) for comparison. As shown in SFig.4, the standardized PDF of SWE among the ensemble members appears to follow a symmetric quasi-Gaussian distribution for both the global and transition regions.

[Figure]

Supplementary Fig. 4 Probability density function (PDF) of the standardized SWE ensemble perturbation, with the ensemble mean removed from each ensemble member, at each grid point. The PDF is averaged globally (blue line) and for the transition region (red line) for April from 2013 to 2020. The black line represents the standardized Gaussian function N(0,1).

**Reply to the Reviewer (#2)′s Comments :**

**I think this version of the manuscript addresses the concerns of the previous reviewers. I have just a few minor suggestions:**

**1) 26 'Snow Water Equivalent (SWE), as one of the land initial conditions, ' -> Initial or boundary conditions**

Response) Modified as suggested.

**2) 33 'This constitutes a novel approach that has not been previously attempted'**
**: This is not totally correct. There are quite a few papers on snow depth or snow water equivalent assimilation. A search in Google Scholar under the search field 'snow water equivalent assimilation' yields several pages of papers since 2020, which are not cited or discussed in the introduction. Perhaps the authors would want to place their study in the context of those papers**

Response) As the reviewer pointed out, while there are existing studies on SWE data assimilation (e.g., Oaida et al., 2019; Smyth et al., 2020; Luojus et al., 2021), the use of passive microwave observations based on the LETKF in this context is relatively rare (e.g., Girotto et al., 2020). Specifically, the approach of optimally combining AMSR2 and IMS satellite data with the JULES LSM model for data assimilation, as done in this study, represents a first attempt and provides a distinctive contribution. In agreement with the reviewer's comment, we have removed the term "novel" from the sentence and revised the sentence accordingly. Additionally, we have included references to existing SWE data assimilation studies in the manuscript.

Revision) (L33-36) This approach constitutes an objective method that optimally combines two previously unattempted incomplete data sources: the satellite SWE retrieval from the Advanced Microwave Scanning Radiometer 2 (AMSR2) and dynamically-balanced SWE from the JULES land surface model.

(L147-160) In previous studies, various approaches have been attempted to improve SWE product performance, such as combining satellite-derived SWE with ground observations (Pulliainen et al., 2020), different satellite data sets (Gan et al., 2021), simple snow models

(Dziubanski and Franz, 2016), or LSMs (Kwon et al., 2017; Kumar et al., 2019). However, most previous studies have focused on targeted regions with limited ground-based observations. Snow initialization in global coverage using satellite-derived SWE remains a persistently challenging task.

Therefore, this study developed an advanced SWE data assimilation framework with satellite remote-sensing data using the local ensemble transform Kalman filter (LETKF) and the Joint U.K. Land Environment Simulator (JULES) land model. While there are existing studies on SWE data assimilation (e.g., Oaida et al., 2019; Smyth et al., 2020; Luojus et al., 2021), the use of passive microwave observations based on the LETKF in this context is relatively rare (e.g., Girotto et al., 2020).

Add reference)

Girotto, M., Musselman, K. N., and Essery, R. L. H.: Data Assimilation Improves Estimates of Climate-Sensitive Seasonal Snow, Current Climate Change Reports, 6, 81–94, https://doi.org/10.1007/s40641-020-00159-7, 2020.

Luojus, K., Pulliainen, J., Takala, M., Lemmetyinen, J., Mortimer, C., Derksen, C., Mudryk, L., Moisander, M., Hiltunen, M., Smolander, T., Ikonen, J., Cohen, J., Salminen, M., Norberg, J., Veijola, K., and Venäläinen, P.: GlobSnow v3.0 Northern Hemisphere snow water equivalent dataset, Sci. Data, 8, 163, https://doi.org/10.1038/s41597-021-00939-2, 2021.

Oaida, C. M., Reager, J. T., Andreadis, K. M., David, C. H., Levoe, S. R., Painter, T. H., Bormann, K. J., Trangsrud, A. R., Girotto, M., and Famiglietti, J. S.: A High-Resolution Data Assimilation Framework for Snow Water Equivalent Estimation across the Western United States and Validation with the Airborne Snow Observatory, J. Hydrometeorol., 20, 357–378, https://doi.org/10.1175/JHM-D-18-0009.1, 2019.

Smyth, E. J., Raleigh, M. S., and Small, E. E.: Improving SWE Estimation With Data Assimilation: The Influence of Snow Depth Observation Timing and Uncertainty, Water Resour. Res., 56, e2019WR026853, https://doi.org/10.1029/2019WR026853, 2020.

**3) 41 'superior performance in high-latitude regions' : superior is not the right word here. Why not use just better ?**

Response) Modified as suggested.

**4) 52 'This assimilation framework is anticipated to contribute to a more precise prediction of atmospheric conditions by realistically capturing the interaction between**

**the atmosphere and land, given the substantial influence of SWE on energy and water balance at the interface of the atmosphere and land'**

**: This paragraph does not fit in the abstract. It is not informative, and, as I mentioned in a previous point, there are many other SWE assimilation schemes.**

Response) We agree with the reviewer's comment and have deleted the corresponding sentence from the abstract.

**5) The methods section and, I believe, the manuscript as a whole do not indicate that the assimilation is done at the grid-cell level. This has some consequences for the assumptions of the observational errors, as the observation error-covariance is assumed to be diagonal.**

Response) Thank you for your comment. Following the reviewer's comment, to prevent any misunderstanding, we have added the following clarification to the manuscript that data assimilation is performed at each grid point:

Revision) (L282-284) To address this issue, we attempted to apply a simple and effective standard normal deviation scaling to satellite-derived SWE at each grid point, considering its potential use as initial conditions for JULES LSM-based climate models.

(L296-298) The snow assimilation is conducted based on the LETKF (e.g., Hunt et al., 2007), which is utilized to combine remotely sensed retrievals with the LSM model outputs (a.k.a. backgrounds) at each grid point to produce a snow analysis.

(L325-327) This study conducts the advanced daily cycle snow data assimilation experiment at each gird point using the LETKF based on the satellite data and the JULES LSM model outputs driven by 3-hourly JRA55 reanalysis atmospheric forcing.

---

## Author Response (AR3)

Response to Reviews of ″Assimilation of snow water equivalent from AMSR2 and IMS satellite data utilizing the local ensemble transform Kalman filter″ by Joonlee lee, Myong-In Lee, Sunlae Tak, Eunkyo Seo, and Yong-Keun Lee. (Geoscientific Model Development: #gmd-2023-221)

We would like to thank the reviewers for their valuable feedback. Their insightful comments helped improve the quality of this paper. After examining the reviewers' comments, we have corrected and modified our manuscript. Our responses to the individual comments are provided below in blue.

**Reply to the Reviewer (#1)′s Comments:**

As this paper has undergone several review iterations and it appears that the authors have implemented the revisions suggested by previous reviewers, I will keep my review brief and limited to the following minor comments and suggestions:

Line 27: "… owing to its enduring memory…". Since "memory" has not yet been defined as intended by the authors up to the third line in the abstract, I suggest revising the wording.

Response) In response to the reviewer's suggestion, we have replaced the term "memory" with "persistence."

Revision) (L25-28) Snow Water Equivalent (SWE), as one of the land initial or boundary conditions, plays a crucial role in global or regional energy and water balance, thereby exerting a considerable impact on seasonal and sub-seasonal scale predictions owing to its enduring persistence over 1 to 2 months.

Line 327: Typo "gird point" should be corrected.

Response) Modified as suggested.

Line 357: Revise the sentence to "differences from IMS for AMSR2, Openloop,…" to be consistent with the figure caption.

Response) Modified as suggested.

**Line 358: Clarify if SCF and SWE are being used interchangeably (Figure 2 displays SCF).**

Response) Figure 2 represents the SCF from JRA-55. We have corrected the typographical error in the sentence by replacing "SWE" with "SCF."

Revision) (L358-360) Here, the JRA55 SCF serves as a reference dataset for comparison with other reanalyses and is associated with meteorological forcing data used in the JULES land surface model.

**Figures: Add color-scale labels (with parameter and its unit) in the figures.**

Response) We have added the color-scale labels' units to the captions of each figure, as requested. SWE is measured in units of $kg/m^2$ or mm, while Spearman rank correlation is dimensionless. Like the correlation, SCF is also dimensionless since it represents a proportion rather than a physical measurement with units.

**Figure captions 2, 3, 4, 5, 6, 7, 10 (and other instances in the paper and in the SI): The statement "Negative values in red shades are indicated with a diagonal line" is unclear as the diagonal lines are not visible. Please clarify their purpose.**

Response) To ensure clarity for readers with color vision deficiencies, hatching has been applied to indicate negative values. Specifically, we have revised Figures 2, 3, 4, 7, 9, 10, and SFig. 3 to distinguish between positive and negative values accordingly.